# Breathing Storms: Enhanced Ecosystem Respiration During Storms in a Heterotrophic Headwater Stream

Carolina Jativa<sup>1</sup>, Anna Lupon<sup>1</sup>, Emma Lannergård<sup>2</sup>, José L. J. Ledesma<sup>3,4</sup>, Gerard Rocher-Ros<sup>5</sup>, Xavier Peñarroya<sup>1</sup>, Susana Bernal<sup>1</sup>

<sup>1</sup>Integrative Freshwater Ecology Group, Centre for Advanced Studies of Blanes (CEAB-CSIC), Blanes, 17300, Spain <sup>2</sup>Department of Aquatic Sciences and Assessment, Swedish University of Agricultural Sciences, Uppsala, 75007, Sweden <sup>3</sup>Department of Biogeochemistry and Microbial Ecology, National Museum of Natural Sciences - Spanish National Research Council (MNCN-CSIC), Madrid, 28006, Spain

<sup>4</sup>Department of Hydrogeology, Helmholtz Centre for Environmental Research - UFZ, Leipzig, 04318, Germany <sup>5</sup>Integrated Science Lab, Department of Ecology, Environment and Geoscience, Umeå University, Umeå, 90187, Sweden *Correspondence to*: Carolina Jativa (carolina.jativa@ceab.csic.es)





**Abstract.** Hydrological disturbances following storm events influence the structure and functioning of headwater streams. However, understanding how these disturbances impact critical processes such as stream metabolism is challenging. We assessed the effect of storm events on the resistance and resilience of gross primary production (GPP) and ecosystem respiration (ER) in a heterotrophic headwater stream. We hypothesized that stream metabolism will show low resistance to storm events because GPP and ER will be either stimulated by inputs of limited resources (small storms) or suppressed by biofilm damage (large storms). We also expected resilience to decrease with the size of the storm event. To test these hypotheses, we hydrologically characterized 53 individual storm events during 4.5 years (period Oct 2018 - Feb 2023) and estimated metabolic rates prior, during, and after each event. Individual storm events had different duration (from 4 to 32 days), and exhibited contrasting changes in discharge (discharge from 0.6 to 872.4 L s<sup>-1</sup>). Due to data and model constraints, we were able to estimate metabolic rates for 35 of these events, for which GPP and ER averaged  $1.7 \pm 1.8$  and  $-13.4 \pm 7$  g  $O_2$ m<sup>-2</sup> d<sup>-1</sup>, respectively. The two processes showed low resistance to storm events, with magnitudes increasing in 69% and 86% of the cases for GPP and ER, respectively. The relationship between hydrological parameters and changes in GPP was not statistically significant, while a positive relation with the magnitude of the storm event was found for ER ( $\mathbb{R}^2 > 0.37$ ). Similarly, recovery times were positively related to the size of the event only for ER ( $R^2 > 0.46$ ). Yet recovery times were always lower than 6 days, suggesting that the positive effect of resource inputs on stream metabolic activity was limited over time. Our findings support the idea that storm events stimulate metabolic activity in headwater streams, especially ER, and highlight how changes in hydrological regimes could impact stream functioning and its role in global biogeochemical cycles.

# 30 1 Introduction

Stream metabolism regulates energy and matter fluxes within running waters, as it includes the processes of carbon fixation (gross primary production, GPP) and mineralisation (ecosystem respiration, ER). GPP and ER are key ecosystem functions as they control organic matter processing, nutrient cycling, greenhouse gas emissions, and aquatic food webs (Roberts et al., 2007). Further, stream metabolism is highly sensitive to both climate and anthropogenic disturbances, making metabolic rates an increasingly valuable tool for understanding aquatic ecosystems' functioning and their response to global change perturbations (Young et al., 2008; Bernhardt et al., 2018; Palmer and Ruhi, 2019).

Stream metabolism is influenced by several abiotic and biotic factors that control its variability over space and time. Light inputs, temperature, and nutrient concentrations are the main drivers of GPP, both across rivers and within the same river over time (Mulholland et al., 2001; Roberts and Mulholland, 2007; Lupon et al., 2016; Savoy et al., 2019), while ER is usually regulated by temperature (Uehlinger, 2000; Acuña et al., 2004) and organic matter availability (Demars, 2019; Lupon et al., 2023). All these factors are directly or indirectly related to stream hydrology, as increases in sediment and solute export during high flows generally alter turbidity, chemistry (Dodds et al., 2002; Webster et al., 2003), and temperature (Butturini and Sabater, 1998; Savoy et al., 2019) of stream water. Hydrology also controls the interaction between available solutes and stream biofilms, primarily through its control over transient storage (Grimm and Fisher, 1984; Hall et al., 2002), water residence time (Valett et al., 1996), and benthic leaf litter distribution (Mulholland et al., 1985). The influence of hydrology on stream metabolism can be especially pronounced in streams with highly variable hydrological regimes, such as intermittent rivers. Understanding how stream flow, especially during storm events, influences metabolic activity is critical to assess ecosystem functioning and its response to flow regulation or changing climatic conditions.

In recent years, high-frequency measurements of dissolved oxygen (DO) concentration have been conducted in diverse fluvial ecosystems in order to assess the effect of discharge and hydrological disturbance on GPP and ER (i.e., Roberts et al., 2007; Gómez-Gener et al., 2020; O'Donnell and Hotchkiss, 2022). For instance, it has been shown that changes in the hydrologic regime highly affect the seasonal patterns of GPP rates, but not its interannual variability (Marzolf et al., 2024). Yet, there is no clear consensus on whether storm events stimulate or suppress metabolic rates, partly because increasing discharge can shift the balance between biological demand and physical transport processes in either direction (Covino, 2017). A storm event may stimulate stream metabolism by supplying limiting resources, such as nutrients and organic matter. Conversely, a storm event can suppress metabolic rates by increasing turbidity and reducing light availability, shortening water residence times, and physically disturbing benthic communities. This potential for contrasting responses resonates with the Intermediate Disturbance Hypothesis (Connell, 1978), which postulates that biological activity is higher when the disturbance is intermediate in frequency and intensity. Supporting this idea, previous studies have related elevated metabolic rates during small to medium-sized storms to increases in carbon and nutrient exports from surrounding terrestrial ecosystems (Jowett and Biggs, 1997; Demars, 2019; Lupon et al., 2019; Li et al., 2024). However, such stimulation may eventually reach an asymptote when the supply of resources surpasses the ecosystem's processing capacity (sensu River Network Saturation concept;

Wollheim et al., 2018). Upon this threshold, one may expect no additional increases in either GPP or ER. Finally, during large storm events, the advective power of water can increase sediment transport and turbidity (Bernhardt et al., 2018), thereby dislodging the substrate and litter from the streambed (Roberts et al., 2007), decreasing mean water residence time, scouring the benthic biomass (Talbot et al., 2018; O'Donnell and Hotchkiss, 2022), and limiting light availability (Hall et al., 2015). These physical and biological disruptions may ultimately reduce in-stream processing and promote the pulse and shunt of carbon and nutrients across the entire river network (Raymond et al., 2016).

The role of storm events as drivers of metabolic stimulation or suppression can be described in terms of the resistance and resilience of stream ecosystems to hydrological disturbances. The resistance to a hydrological disturbance can be defined as the ability of stream ecosystems to uphold metabolic rates unchanged relative to base flow conditions despite increases in discharge. Hence, either the stimulation or suppression of metabolic activity during storms indicates low resistance to storm perturbations (i.e., Roberts et al., 2007; Uehlinger, 2000; O'Donnell and Hotchkiss, 2022). In turn, the resilience of stream metabolic activity to hydrological events is defined by the time required for either GPP or ER to return to values comparable to those prior to the storm event (Carpenter et al., 1992; O'Donnell and Hotchkiss, 2022). Streams with high resilience are those for which metabolic rates quickly recover prior values after the storm event. The duration of this recovery can vary depending on the magnitude of the disturbance, as well as across seasons and on environmental conditions such as light availability and temperature (Roberts et al., 2007; Lowman et al., 2024).

The objective of this study was to assess the response of stream metabolism to storm events of different magnitude, intensity, and duration. To do so, we examined the hydrological characteristics and metabolic response of 53 individual storms in a non-perennial, forested headwater stream showing large temporal variability in discharge (Bernal et al., 2002, 2019). We hypothesized that ER would show low resistance (i.e., abrupt changes) to storm events, by either increasing when intermediate events alleviate resource limitation (metabolic stimulation) or decreasing when large events damage biofilm and promote the pulse and shunt of carbon (metabolic suppression). By contrast, we expected that GPP would generally exhibit high resistance because it would be limited by light availability in this closed-canopy stream (Fig. 1a). Only during periods of open-canopy (i.e., early spring and late fall), GPP would behave similarly to ER. Further, we hypothesized that stream metabolism would show lower resilience (i.e., longer recovery periods) with increasing magnitude and duration of the storm due to higher metabolic changes and longer periods of land-water connectivity. Yet, we anticipated that recovery time may reach a threshold corresponding to the time required for biofilms to rebuild after large storm disturbances (Fig. 1b).

Figure 1: Conceptualization of the impact of storm magnitude on the (a) resistance and (b) resilience of stream metabolic rates. We hypothesized that, under no light limitation (dark blue line), stream metabolism has low resistance to hydrological disturbances. It is initially stimulated during small to intermediate events and then suppressed during large events. Further, stream metabolic rates have low resilience to hydrological disturbances, showing prolonged recovery times during larger events until reaching a threshold corresponding to biofilm damage. Under light-limited conditions (orange line), we hypothesized that ER has the same behaviour described above, while GPP shows higher resistance and resilience regardless of the storm magnitude until reaching the threshold of biofilm damage.

# 2 Methods

# 100 **2.1 Study Site**




The study was conducted in Fuirosos, a three-order stream located in the Montnegre-Corredor Natural Park in Northeast Spain (41° 41′ 04.5″ N, 2° 34′ 46.0″ E). The climate is Mediterranean, with warm, dry summers and mild, humid winters. Monthly mean temperatures range from 3°C in January to 24°C in August. Mean annual precipitation is  $658 \pm 216$  mm (mean  $\pm$  standard deviation). Most precipitation falls during spring and autumn, with sporadic storms in summer and winter.

The drainage area at the catchment outlet is 18.7 km² and the channel width ranges between 3 to 5 meters. The catchment lies on a fine-grained granitoid batholith, resulting in sandy and poorly developed soils. The landscape is predominantly forested, with perennial forests of *Quercus suber, Quercus ilex, Pinus pinea, and Pinus halepensis* covering over 85% of the catchment area. The population density is < 5 inhabitants per km², which can be perceived as minimal with regard to human interference. The stream is flanked at the valley by a well-developed riparian zone occupying 6% of the catchment area. The main riparian species are *Alnus glutinosa* and *Platanus acerifolia*.

Fuirosos is a non-perennial stream primarily fed by springs and groundwater inputs. As most Mediterranean non-perennial streams, the flow regime is extremely erratic (Bernal et al., 2019), with stream discharge ranging from 0 to 3884 L s<sup>-1</sup> and an average water velocity of  $0.11 \pm 0.2$  m s<sup>-1</sup> (period October 2018 to February 2023). The lateral connectivity between the stream and the surrounding riparian zone also has a marked seasonal pattern, as it switches from being a gaining stream during the wet period (November-April) to a losing stream during the dry season (May-November) (Butturini et al., 2003). Further, stream flow usually ceases for several weeks in summer and resumes during autumn storm events. Isolated pools can persist for several days until the stream dries completely (Butturini et al., 2002). In our study, we only considered storm events that took place during periods with surface flow.

The Fuirosos stream is relatively oligotrophic. Long-term dissolved organic carbon (DOC) concentrations are generally low, averaging around 4 mg C L<sup>-1</sup>. Yet they experience a significant rise in autumn, reaching up to 8 mg C L<sup>-1</sup> (Vazquez et al., 2011). Long-term stream nitrate (NO<sub>3</sub><sup>-</sup>) concentration averages  $210 \pm 50 \mu g$  N-NO<sub>3</sub><sup>-</sup> L<sup>-1</sup>, with the highest concentration typically observed in winter (Bernal et al., 2005).

#### 2.2 Data collection

Meteorological data, including daily precipitation, were obtained from the Dosrius station managed by the Catalan Meteorological Service and located ca. 15 km southwest of the outlet of Fuirosos (SMC, 2023). The suitability of this station for representing conditions in Fuirosos was carefully evaluated and confirmed in Ledesma et al. (2021).

In order to estimate stream metabolism, in October 2018 we installed a monitoring station in the stream with an upslope contributing area of 9.9 km<sup>2</sup>. The 200-m section upstream of the sampling station was bordered by a well-developed riparian

forest. Further, this section was minimally influenced by tributaries, had a slope of 1.9%, and was representative of the stream hydromorphology featuring small runs, riffles, and a few shallow pools. During the study period, the average water depth was  $7.5 \pm 8$  cm and wetted width was  $2.4 \pm 0.5$  m. The monitoring station was equipped with a suite of sensors that measured both physical (incident light, water depth, and water temperature) and chemical (dissolved oxygen) properties at a 10-minute frequency. Incident light (lux) was recorded every 10 minutes using two HOBO UA-002-64 loggers (Onset Corporation) installed in riparian trees near the sampling station. Lux values from the two sensors were averaged and converted to photosynthetic photon flux density (PPFD, in umol m<sup>-2</sup> s<sup>-1</sup>) using a conversion factor of 0.0185, which is commonly applied for forested environments (Thimijan and Heins, 1983). The resulting PPFD values were then aggregated into daily totals. A threshold of 4 mol m<sup>-2</sup> d<sup>-1</sup> was considered to be the minimum PPFD required to support stream photoautotrophic activity (Hill et al., 1995). Stream water depth (h, in cm) was estimated from 10-minute pressure measurements using a HOBO water level logger installed in a stilling well, with atmospheric pressure corrections applied using a paired barometric logger located in a nearby tree. To verify the representativeness of these data across the 200-m study reach, manual depth measurements were conducted biweekly at randomly selected transects. These manual measurements were also used to calibrate the water level data, confirming that the sensor reliably captured overall depth variability during both baseflow and storm conditions. Water level measurements were converted to instantaneous stream discharge (Q, L s<sup>-1</sup>) using pre-established rating curves, which were developed from reconstructed sensor stream water levels and field-measured Q taken ca. biweekly (n = 50). Finally, water temperature (T. in °C) and DO concentrations (in mg L<sup>-1</sup>) were measured with a MiniDOT logger (PME, USA), which employs a fluorescent optode sensor known for long-term stability and robustness in field deployments. The logger was firmly anchored next to the water depth sensor with a perforated plastic pipe as a protective casing. From October 2018 to February 2023, all sensors were checked out every 15-20 days for maintenance and data downloading. Although the MiniDOT generally requires little maintenance, the sensing surface and housing were gently rinsed with stream water when needed to limit biofouling.

#### 2.3 Storm event characterization








To identify and characterize individual storm events from the hydrograph, we used the method described by Lannergård et al. (2021). Briefly, the start of an event was defined as the moment when daily discharge increased for two consecutive days at a rate exceeding 1.5%. An individual event was considered to end when either a new event began or when the Q dropped below the level predicted by a first-order baseflow decay function, which was calculated using Q at the start of the event and assumed to decrease by 0.99% per day. Additionally, if the decay function was met but Q continued to decrease by more than 0.05%, the event continued until there was no further decrease in Q. We only kept those events following accumulated precipitation higher than 10 mm, which is the minimum amount of precipitation needed to generate a hydrological response in this stream (Bernal et al., 2002). By following this procedure, we were able to identify 53 individual storm events for the study period (Table S1).

For each individual event, we extracted the following variables: total precipitation (P, in mm), maximum precipitation intensity (PI<sub>max</sub>, in mm  $h^{-1}$ ), duration of the storm event (D, in days), runoff coefficient (RC, in %), and change in discharge ( $\Delta Q$ , in L s<sup>-1</sup>). The P, PI<sub>max</sub>, and D provide insights into the size of the event. The RC was calculated using the total water flux in mm during each storm event, estimated as the integral of Q over P. This cumulative approach accounts for antecedent moisture conditions and captures the stream's overall hydrological response to each storm event, avoiding dependence on potentially unrepresentative single-point Q values (Bernal et al., 2002). Finally, the  $\Delta Q$  is the difference between the peak flow and the average discharge during the three days before the event ( $Q_{prior}$ ) and indicates the magnitude of the disturbance caused by the storm event.

### 2.4 Metabolism calculations





We estimated daily stream metabolic rates during each storm event as well as for the prior and following weeks. GPP and ER (in g O<sub>2</sub> m<sup>-2</sup> d<sup>-1</sup>) were estimated using the Bayesian inverse model *b\_Kb\_oipi\_tr\_plrckm.stan* from the *streamMetabolizer* R package (version: 0.12.1), which incorporates both process and observation errors (Appling et al., 2018) (Appendix 1, Table S2). We assumed that GPP is a linear function of light intensity (Van de Bogert et al., 2007), while ER is constant throughout the day. The model does not account for factors that could increase ER during the day (Hotchkiss and Hall, 2014), such as photorespiration, which is presumed to be minimal in our forested headwater stream (Parkhill and Gulliver, 1998).

All inputs for the *streamMetabolizer* model (i.e., light, water temperature, depth, DO concentration, and DO saturation) were derived from the high-frequency sensor data described in Section 2.2. DO saturation was calculated from 10-minute water temperature and barometric pressure values using the García and Gordon (1992) solubility equation. Sensor data were quality-checked by removing implausible values (e.g., negative DO concentration, sensor spikes) and smoothed using LOESS regression (span = 0.03) to reduce noise and filter outliers. Details of the data preparation procedures are provided in Appendix 1 in the supplementary materials.

Following the procedure by Bernal et al. (2022), GPP, ER, and the gas transfer coefficient ( $K_{600}$ , in d<sup>-1</sup>) were fitted daily based on DO dynamics over a 24-hour period starting at 23:00h the previous day. Prior probability distributions for GPP and ER were set based on previously reported values ( $0.5 \pm 10$  and  $-5 \pm 10$  g O<sub>2</sub> m<sup>-2</sup>d<sup>-1</sup> for GPP and ER, respectively) (Acuña et al., 2004; Bernal et al., 2022). Prior probability of  $K_{600}$  was strongly constrained to minimize the problem of equifinality (Appling et al., 2018) (Fig. S2). To constrain  $K_{600}$ , we run the Bayesian model using a deterministic, hand-pooled  $K_{600}$  obtained from the relationship between binned Q and the  $K_{600}$  estimated from a series of 11 propane additions using a mixed tracer injection method (Jin et al., 2012) (Fig. S1a). Additionally, we verified the feasibility of the modelled  $K_{600}$  using independent  $K_{600}$  predictions from both the night-time regression method (Odum, 1956) and hydraulic geometry (Raymond et al., 2012) (Fig. S1b). Only estimates of GPP and ER that passed a quality check and satisfied all the criteria were included. Days with biologically or physically implausible values (i.e., GPP < 0, ER > 0,  $K_{600}$  > 110 d<sup>-1</sup>), poor model convergence (i.e., R-hat > 1.2 and number of effective samples > 8000), or poor fit to DO data (i.e.,  $R^2$  < 0.50, root mean square error (RMSE) > 0.4, mean

absolute error (MAE) > 0.4) were excluded. This procedure allowed us to calculate the metabolic rates for 66% of storm events (35 out of 53 events). Of the 18 excluded events, six lacked complete DO data, and 12 did not pass the mentioned quality test. Notably, all days with  $Q > 100 \text{ L s}^{-1}$  failed the quality test, leading to the exclusion of metabolic rate calculations during large storm events (Table S3). Consequently, we were unable to test our expectation that metabolic rates would decrease during large storm events.

For each storm event, we assessed the resistance of metabolic activity ( $\Delta$ MET) as the relative change in either GPP ( $\Delta$ GPP) or ER ( $\Delta$ ER) as follows:

$$\Delta MET = \frac{|MET|_{Peak} - |MET|_{Prior}}{|MET|_{Prior}} x \ 100\%,$$




where  $\Delta$ MET is the percentage of change of either GPP or ER, and |MET|<sub>Peak</sub> (in absolute values) is either the maximum (when there is stimulation) or the minimum (when there is suppression) metabolic rate observed during the event. |MET|<sub>Prior</sub> is the average metabolic rate (in absolute values) during the three days preceding the event. For |MET|<sub>Peak</sub>, we used the maximum (or minimum) metabolic rate rather than the metabolic rate at the peak discharge because there was typically a two-day gap between the discharge and metabolic peaks. We considered that  $\Delta$ MET values higher than +10% indicate a stimulation of the metabolic process, whereas values of  $\Delta$ MET lower than -10% indicate a suppression of metabolic rates. Finally, we considered  $\Delta$ GPP or  $\Delta$ ER values within  $\pm$ 10% to indicate stable metabolic rates, and hence, high resistance of stream metabolism to storms.

To assess the resilience of metabolic rates, we calculated the recovery time (RT, in days) of GPP and ER for each storm event. Recovery time was determined as the number of days required for either GPP or ER to return to conditions similar to MET<sub>Prior</sub> (i.e., within  $\pm$  10%). When there was insufficient data during the recession limb or when recovery time overlapped with another event (n = 7), we inferred the day when either GPP or ER returned to pre-event values from a linear regression model between time and metabolic rates during the recession limb of the hydrograph.

# 2.5 Statistical analysis

We used linear and logarithmic regression models to explore the relationships between (i) hydrological descriptors (PI<sub>max</sub>, D, RC, and ΔQ), physical parameters (daily PAR and mean daily T), and metabolic rates (GPP, ER) during storm events, (ii) hydrological descriptors and both metabolic resistance (assessed as metabolic change; ΔGPP and ΔER) and resilience (assessed as recovery time; RT<sub>GPP</sub> and RT<sub>ER</sub>), and (iii) metabolic resistance and resilience for GPP and ER. We selected the best-fit model using Akaike Information Criterion (AIC) (Akaike, 1974). The two models were considered equally good when the differences in AIC were less than 2. The goodness of fit for each regression model was assessed using the coefficient of determination (R²). Regressions were considered marginal and strong statistically significant when p-values were lower than 0.05 and 0.01, respectively. All data analyses were conducted using R software (v. 4.3.3).

#### 3 Results





# 3.1 Characterization of storm events

During the study period, both meteorological and hydrological variables varied across seasons and hydrological years (Fig. 2). Annual precipitation ranged from 428 mm in 2019 to 1,172 m in 2020, and mean annual temperature ranged from 13.8 °C in 2021 to 15.2 °C in 2022. Further, mean daily Q spanned from  $5.67 \pm 21.9$  L s<sup>-1</sup> (mean  $\pm$  SD) in 2021 to  $59.28 \pm 317.9$  L s<sup>-1</sup> in 2020. The duration of the summer dry periods was also variable: there were no dry days in 2020, while 2019 had 115 days without water flow (Fig. 2b). The hydrological descriptors (P, PI<sub>max</sub>, D, RC, and  $\Delta$ Q) covered a wide range of values, most of them expanding at least one order of magnitude among individual storm events (light blue bars, Fig. 3). P ranged from 8.6 to 234.9 mm (46  $\pm$  5.7 mm), PI<sub>max</sub> ranged from 1.30 to 31.4 mm h<sup>-1</sup> (10.8  $\pm$  0.98 mm h<sup>-1</sup>), and D ranged from 4 to 32 d (13  $\pm$  7 d) (Fig. 3a, b). The magnitude of the storm event also differed widely among events: Q<sub>prior</sub> ranged from 0.7 to 165 L s<sup>-1</sup> (16.5  $\pm$  3.9 L s<sup>-1</sup>), while  $\Delta$ Q ranged from 0.6 to 3,501 L s<sup>-1</sup> (150  $\pm$  70 L s<sup>-1</sup>). Finally, RC ranged from 0.22 to 39.44% (7.2  $\pm$  1.9%) (Fig. 3c, d). Light and water temperature also varied across individual storm events. Daily PAR ranged from 2 to 28.6 mol m<sup>2</sup> d<sup>-1</sup> and was lower than the threshold to support stream photoautotrophic activity for 19 of the 53 cases. Mean daily water temperature varied from 4.3 to 21 °C.

Figure 2: Temporal variation of a) daily precipitation (P), b) mean daily discharge (Q), and c) mean daily dissolved oxygen concentrations (DO) at the Fuirosos stream from October 2018 to February 2023. Dashed vertical lines represent each of the 53 identified storm events. Yellow shaded areas represent periods with no flow.

Figure 3. Histograms showing the distribution of (a) maximum precipitation intensity ( $PI_{max}$ ) (b) duration of storm event (D), (c) change in discharge ( $\Delta Q$ ), and (d) runoff coefficient (RC) for the 53 identified storm events (light and dark blue) as well as for the 35 events for which we could calculate metabolic rates (dark blue).

#### 3.2 Metabolic rates during storm events




From the 53 storm events identified, we were able to calculate metabolic rates for 35 events that expanded across most of the range covered by hydrological and environmental descriptors (dark blue bars, Fig. 3). During these storm events, GPP ranged from 0.005 to 10.6 g  $O_2$  m<sup>-2</sup> d<sup>-1</sup> (1.7 ± 1.8g  $O_2$  m<sup>-2</sup> d<sup>-1</sup>) and showed a marginal linear relationship with PI<sub>max</sub>, Q, and light (p < 0.05), though the variance explained was low in all cases (R<sup>2</sup> < 0.13). GPP was not related to D, RC, or temperature (in all cases; p > 0.01) (Table S4). Rates of ER were 9 times higher than GPP, ranging from -4.3 to -36.4 g  $O_2$  m<sup>-2</sup> d<sup>-1</sup> (-13.4 ± 7 g  $O_2$  m<sup>-2</sup> d<sup>-1</sup>). ER was positively related to Q (linear regression, R<sup>2</sup> = 0.21, p < 0.0001) (Fig. 4b). ER and stream temperature had a weak positive relationship, yet the variance explained was minimal (R<sup>2</sup> = 0.06, p < 0.001). There was a marginal positive relationship between ER and PI<sub>max</sub>, though the variance explained was low (R<sup>2</sup> = 0.17, p 

Figure 4. Relationship between mean daily discharge (Q) and daily a) gross primary production (GPP) and b) ecosystem respiration (ER) across all individual storm events (in blue) and for the days 1 to 6 prior to each event (orange). The black line represents the linear regression between variables (only shown when p < 0.01) and the shaded area indicates the 95% confidence interval. Note that regressions were based only on storm event days.

#### 3.3 Response of metabolic rates to the storm events



Stream metabolic rates changed in response to the storm events, but the proportion of cases showing stimulation, suppression, or no change differed between GPP and ER. Stimulation was observed during 69% and 86% of the cases (24 and 30 out of 35 cases) for GPP and ER, respectively, whereas suppression was observed more often for GPP (9 out of 35 cases) than for ER (1 out of 35 cases) (Table 1). Only 6% of the events (2 out of 35) showed no change in GPP, while no changes in ER occurred in 11% of the events (4 out of 35) (Table 1).

| Event | GPP                       | ER                    | Δ GPP   | $RT_{GPP}$ | ΔER    | $RT_{ER}$ |
|-------|---------------------------|-----------------------|---------|------------|--------|-----------|
|       | g O <sub>2</sub> m -2 d-1 | $g O_2 m^{-2} d^{-1}$ | %       | d          | %      | d         |
| 2     | $0.5 \pm 0.3$             | $-12.8 \pm 1.6$       | + 100.8 | 2          | + 24.4 | 4         |
| 4     | $0.7 \pm 0.1$             | $-14.5 \pm 1.9$       | + 29    | 2          | +33.1  | 4         |
| 7     | $1.2 \pm 0.1$             | $-10.4 \pm 1.2$       | 0       | 0          | + 10.4 | 1         |
| 8     | $1.6 \pm 0.3$             | $-6.74 \pm 0.6$       | +20.9   | 1          | + 16.2 | 1         |
| 9     | $1.6 \pm 0.5$             | $-7.51 \pm 2.1$       | +26.5   | 2          | +37.2  | 2         |
| 10    | $1.3 \pm 0.5$             | $-7.44 \pm 1.5$       | + 67.9  | 4          | +62.3  | 3         |
| 11    | $0.8 \pm 0.1$             | $-6.86 \pm 0.7$       | + 17.9  | 1          | +32.5  | 1         |
| 12    | $1.1 \pm 0.4$             | $-6.96 \pm 1.2$       | + 62.4  | 1          | +23.6  | 1         |
| 13    | $0.7 \pm 0.6$             | $-7.64 \pm 0.6$       | +22.5   | 1          | + 13.3 | 1         |
| 20    | $1.8 \pm 0.5$             | $-17.2 \pm 5.1$       | -76.1   | 2          | +86.9  | 3*        |
| 21    | $1.8 \pm 0.2$             | $-21.5 \pm 2.2$       | -14.8   | 2          | +66.6  | 6*        |
| 22    | $1.5 \pm 0.2$             | $-23 \pm 4.1$         | 0       | 0          | +68.5  | 5         |
| 25    | $0.9 \pm 0.2$             | $-13.4 \pm 1$         | + 91    | 3          | 0      | 0         |
| 26    | $1.1 \pm 0.7$             | $-18.2 \pm 1$         | +48.3   | 2          | 0      | 0         |
| 27    | $1.5 \pm 0.3$             | $-18.5 \pm 1$         | + 38.9  | 1          | 0      | 0         |
| 28    | $2.4 \pm 1.4$             | $-21.9 \pm 2.8$       | +67.1   | 2          | +60.6  | 3         |
| 29    | $4.3 \pm 2.3$             | $-24.9 \pm 4$         | -73.1   | 2          | +30.6  | 1         |
| 30    | $5.9 \pm 2.1$             | $-28.9 \pm 1.7$       | + 15.2  | 2          | + 18.3 | 2         |
| 34    | $0.3 \pm 0.2$             | $-12.9 \pm 7.4$       | +35.8   | 2          | + 85   | 6         |
| 35    | $0.3 \pm 0.1$             | $-10.5 \pm 1.5$       | +85.2   | 5          | +43.5  | 4         |
| 36    | $0.5 \pm 0.2$             | $-5.83 \pm 0.4$       | -87.3   | 1          | +20.9  | 2         |
| 37    | $1.5 \pm 0.5$             | $-11.9 \pm 1.3$       | + 26.6  | 1          | +70.5  | 2*        |
| 38    | $3.5 \pm 1.1$             | $-12.9 \pm 1.1$       | + 141.7 | 8          | +29.2  | 2         |
| 39    | $6.3 \pm 2$               | $-12.5 \pm 1.6$       | +130.3  | 9          | +36.8  | 4         |
| 40    | $2.7 \pm 0.4$             | $-12.2 \pm 2.2$       | +29.7   | 1          | +37.4  | 2         |
| 41    | $1.6 \pm 0.6$             | $-11.1 \pm 0.9$       | -56.2   | 2          | 0      | 0         |
| 42    | $0.1 \pm 0.1$             | $-9.7 \pm 3.7$        | +28.1   | 1          | +51.2  | 4         |
| 43    | $0.1 \pm 0$               | $-7.39 \pm 1$         | -16.1   | 2          | +33.3  | 4         |
| 44    | $0.1 \pm 0$               | $-6.14 \pm 2.9$       | -95.9   | 4          | +39.6  | 5         |
| 45    | $1.4 \pm 1.2$             | $-23.5 \pm 5.6$       | -67.8   | 6*         | + 106  | 6*        |
| 47    | $2.3 \pm 0.6$             | $-16.3 \pm 4.5$       | -21.3   | 2          | + 91.3 | 6         |
| 49    | $0.6 \pm 0.7$             | $-12.4 \pm 1.7$       | + 110.8 | 1          | + 34   | 1         |
| 51    | $0.6 \pm 0.2$             | $-24.7 \pm 3.5$       | +36.8   | 2          | -29.4  | 4         |
| 52    | $0.8 \pm 0.2$             | $-28.5 \pm 3.4$       | +76.6   | 4*         | +47.6  | 4*        |
| 53    | $0.9 \pm 0.1$             | $-6.4 \pm 1.3$        | + 53    | 4          | + 51.4 | 5         |


The change in metabolic rates ( $\Delta$ GPP and  $\Delta$ ER) ranged from -95.9% to +141.7% for GPP, and from -29.4% to +106% for ER, respectively (Table 1). Values of  $\Delta$ GPP were not related to most hydrological descriptors (i.e., D, PI<sub>max</sub>, Q<sub>prior</sub> or RC; in all cases p > 0.01 for either linear or logarithmic relationship). There was a marginal positive linear relationship between GPP and  $\Delta$ Q, but the variance explained was low ( $R^2 = 0.13$ , p < 0.05) (Table S4). In its turn,  $\Delta$ ER exhibited a positive relationship with  $\Delta$ Q, with both linear and logarithmic models providing equally strong fits (Im model:  $R^2 = 0.37$ , p < 0.0001, AIC=325; log model:  $R^2 = 0.39$ , p < 0.0001, AIC=325). ER never doubled relative to prior baseflow levels ( $\Delta$ ER < 106%) (Fig. 5b). Further, all cases showing ER suppression or no change ( $\Delta$ ER  $\leq$  0) were associated with small changes in discharge ( $\Delta$ Q < 10 L s<sup>-1</sup>), whereas ER stimulation ( $\Delta$ ER > 0) occurred across a broader range of  $\Delta$ Q (from 0.6 to 95.6 L s<sup>-1</sup>).


The recovery time (RT) was similarly fast for both metabolic rates,  $2.6 \pm 2$  days and  $3 \pm 2$  days for GPP and ER, respectively (Table 1); and there was no relationship between RT<sub>GPP</sub> and RT<sub>ER</sub> (p > 0.05). As observed for the resistance parameters, RT<sub>GPP</sub> and RT<sub>ER</sub> were not related to either D, PI<sub>max</sub>, or RC (in all cases, p > 0.05 for either linear or logarithmic regressions) (Table S4). Further, there was no relation between RT<sub>GPP</sub> and  $\Delta Q$  (Fig. 5c), while RT<sub>ER</sub> showed a positive relationship with  $\Delta Q$  (Fig. 5d), for which both linear and logarithmic models showed similar goodness of fit (lm model: R² = 0.49, p < 0.0001, AIC=126; log model: R² = 0.46, p < 0.0001, AIC=128). Values of RT<sub>ER</sub> never exceeded the threshold of 6 days. Finally, there was no relationship between  $\Delta$ GPP and RT<sub>GPP</sub> (p > 0.05 for either linear or logarithmic regressions), while there was a moderate and positive linear relationship between  $\Delta$ ER and RT<sub>ER</sub> (R² = 0.45, p 

Figure 5. Relationship between changes in discharge ( $\Delta Q$ , defined as the difference between peak flow and baseflow) and stream metabolic responses. Panels (a) and (b) show the change in gross primary production ( $\Delta GPP$ ) and ecosystem respiration ( $\Delta ER$ ), respectively, used as proxies for metabolic resistance. Panels (c) and (d) show recovery time for GPP ( $RT_{GPP}$ ) and ER ( $RT_{ER}$ ), used as proxies for metabolic resilience. Colour coding indicates whether metabolic activity was stimulated (blue), suppressed (red), or showed no significant change (yellow) in response to the storm event. Solid and dashed lines represent the best-fit models, only when statistically significant (p 

Figure 6. Relationship between resistance and resilience of gross primary production (GPP) and (b) ecosystem respiration (ER) during storm events. Resistance is the relative change in metabolic rates compared to prior baseflow conditions (i.e.,  $\Delta$ GPP and  $\Delta$ ER). Resilience is the recovery time of metabolic rates to prior conditions (i.e.,  $RT_{GPP}$  and  $RT_{ER}$ ). Colour coding indicates whether metabolic activity was stimulated (blue), suppressed (red), or showed no significant change (yellow) in response to the storm event. The black line represents the linear regression between variables (only shown when p < 0.01) and the shaded area indicates the 95% confidence interval.

#### 4 Discussion



In this study, we monitored the hydrology and metabolic activity across 35 individual storm events to understand how stream processes respond to hydrological perturbations in a relatively oligotrophic, heterotrophic, non-perennial headwater stream. The magnitude, intensity and duration of storm events showed substantial variability, with all hydrological descriptors fluctuating by an order of magnitude. For example, the increase in discharge ( $\Delta Q$ ) during the storm events ranged from 0.6 to 872.4 L s<sup>-1</sup>, capturing a broad spectrum of flow hydrographs. Similarly, RC values varied from 0.2% to 39%, which likely

relates to large variability in antecedent soil moisture, land-water hydrological connectivity, and the associated solute supply from terrestrial ecosystems.







Our results showed just a marginal linear relationship between GPP and either discharge or light irradiation. On average, GPP was relatively low  $(1.7 \pm 1.8 \text{ g O}_2 \text{ m}^{-2} \text{ d}^{-1})$ , with values falling at the lower end of the range reported for similarly sized streams (from 0.10 to 22 g O<sub>2</sub> m<sup>-2</sup> d<sup>-1</sup>) (Roberts and Mulholland, 2007; Hall et al., 2016; Savoy et al., 2019). Yet, our estimates are within the range previously reported in this stream (from 0.05 to 1.9 g O<sub>2</sub> m<sup>-2</sup> d<sup>-1</sup>; Acuña et al., 2004, Bernal et al., 2022). Compared to GPP, estimates of ER during the study period were an order of magnitude higher (-13.4 ± 7 g O<sub>2</sub> m<sup>-2</sup> d<sup>-1</sup>) and similar to previous studies (Acuña et al., 2004), highlighting the heterotrophic nature of this stream. ER showed a positive relationship with water temperature, a major driver of metabolic activity, however, the variance explained was minimal. This result might be because water temperature consistently remained above the threshold for sustaining biological activity (> 4 °C, De Nicola, 1996), or because other factors act as primary controls of in-stream heterotrophic activity. In this sense, ER was positively related to stream discharge, which aligns with previous studies pointing out that hydrology is a key driver of stream heterotrophic activity (Roberts et al., 2007; Hall et al., 2017). High flows can enhance heterotrophic activity by promoting the interaction of water with hyporheic sediments, by supplying soil heterotrophic bacteria (Roberts et al., 2007; Li et al., 2021), and, more importantly, by delivering limiting nutrients and organic matter subsidies to the stream (Hinton et al., 1997; Demars, 2019; Lupon et al., 2023). In Fuirosos, this latter explanation is supported by the overall increase in dissolved inorganic nitrogen (DIN) observed during storm flows (Bernal et al., 2002; 2005), which can decrease stream water C:N ratios and meet heterotrophic microbial stoichiometric needs as observed in other Mediterranean streams (Ledesma et al., 2022). This stimulation response of ER is likely enhanced in relatively oligotrophic streams such as Fuirosos, typically exhibiting low DIN concentrations (< 0.6 mg N L<sup>-1</sup>) as well as limited in-stream N uptake rates during base flow conditions (Bernal et al., 2005; Peipoch et al., 2016; Peñarroya et al., 2022).

During individual storm events, both GPP and ER exhibited low resistance, regardless of the magnitude of the storm. Low resistance in GPP was observed in 94% of storm events, with magnitudes changing between -95.9 and 141.7%. Interestingly, 69% of the storm events led to increases in GPP, suggesting that hydrological disturbances can stimulate photoautotrophic activity. These findings agree with previous studies, which attributed this stimulation to a "cleaning effect" of the streambed induced by high flows, when benthic leaf litter and fine particulate organic matter are scored downstream, potentially enhancing light inputs and photosynthetic efficiency (e.g., Roberts et al., 2007; Demars, 2019). However, and contrary to our expectations, GPP stimulation was only marginally related to storm magnitude. This result could be explained by the small photoautotrophic activity measured in this heterotrophic stream, which led to marginal increases in GPP during individual storm events.

We also observed that storms generally stimulate heterotrophic metabolism, as 86% of events showed increases in ER rates compared to base flow conditions. The significant relation between  $\Delta Q$  and  $\Delta ER$  further suggests that greater solute flux

mobilized during storms amplified stream metabolic activity. As seen in previous studies, in Fuirosos, storm events can increase DOC and NO<sub>3</sub> concentrations by 35% and 28%, respectively, further supporting the idea that floods act as subsidies for heterotrophic microbial activity (Butturini et al., 2008). Moreover, the low hydrological connectivity of intermittent streams, driven by periodic disconnection from potential sources (e.g., riparian soils), reduces nutrient inputs from terrestrial ecosystems and exacerbates resource limitation (Bernal et al., 2013), likely explaining the low resistance and high stimulation of ER observed during storms. Conversely, our results suggest a minor effect of antecedent moisture conditions on ER, probably because we were only able to estimate metabolic rates under low RC values (in all cases, RC < 15%), Moreover, ER generally peaked 1–2 days after the discharge peak, which is consistent with prior studies showing that the highest metabolic and nutrient uptake rates usually occur during the beginning of the recession limb of the hydrograph (Roberts et al., 2007; Seybold and McGlynn, 2017). This delay may reflect either a physiological response (i.e., the time required for biofilms to respond to increased resource availability) or a hydrological response (i.e., the lag in the arrival of limiting nutrients from catchment sources). Aside from  $\Delta O$ , we found no relation between changes in metabolic rates (i.e.,  $\Delta GPP$  and  $\Delta ER$ ) and storm duration or intensity. This finding could be explained by the intricate water flow paths and erratic hydrological response to precipitation events exhibited by this intermittent stream (Butturini et al., 2002). Overall, our findings highlight the complexity of biotic responses to hydrological perturbations, which cannot be easily linked to specific storm characteristics such as duration, intensity, or antecedent moisture conditions.








Only one small storm event resulted in ER suppression, which contrasts with previous studies reporting almost no change or mostly suppression of ER during storm events (Reisinger et al., 2017; O'Donnell and Hotchkiss, 2022). O'Donnell and Hotchkiss (2022) explored different magnitudes of storm events, though in a suburban stream with no nutrient limitation, while Reisinger et al. (2017) focused on extreme events that had a destructive effect on stream biofilm. The increase in sediment transport and turbidity (Bernhardt et al., 2018) as well as the reduction in water residence time (Fisher and Grimm, 1988; Uehlinger, 1991) can also contribute to decreased stream metabolic activity during large storm events. In Fuirosos, only 26 days showed discharge exceeding 100 L s<sup>-1</sup>, these high-flow days represented less than 5% of all the storm days analysed, and occurred only during 8 of the 53 identified storm events. Fuirosos is a small stream (median  $Q = 12 \text{ L s}^{-1}$ ; median depth = 7.5 cm), where floods exceeding 100 L s<sup>-1</sup> often lead to overbank flooding and displacement or burial of sensors. Unfortunately, these conditions can disrupt DO signals and complicate the inference of depth and gas exchange, ultimately violating key assumptions of metabolic models. This is likely why none of these 26 days passed quality checks required for reliable metabolic estimates, which preclude us from directly testing the hypothesis that large storms suppress metabolic activity. To further explore this idea, we recalculated metabolic rates during high-flow days (range: 100 to 3884 L s<sup>-1</sup>) after constraining K600 to 100 d<sup>-1</sup>, the most reliable value to occur during high flow conditions based on our expert knowledge (Table S5). This exercise yielded reliable estimates of metabolic rates for only 8 of the 26 high flow days, for which ER values (range: -20 to -41g O<sub>2</sub>  $m^{-2} d^{-1}$ ) were as high as during days with lower  $\Delta Q$  increases (Fig. 4). Therefore, extremely large storms potentially inhibiting ER might be rare in this stream, which more commonly exhibits ER stimulation during storms. Moreover, the relationship between  $\Delta$ ER and  $\Delta$ Q was equally well explained by both linear and logarithmic models, indicating no statistically supported evidence for a saturation pattern in ER. This finding contrasts with the saturation responses often reported for nutrient uptake with increasing discharge (*sensu* River Network Saturation concept; Wollheim et al., 2018). Given that the largest analysed storms fell within a moderate  $\Delta$ Q range compared to extreme hydrological events in similar systems, it is possible that the ecosystem's processing capacity was not fully exceeded, and thus the available range was insufficient to reveal a true saturation response. Nevertheless, we observed apparent constraints in ER during storms. For instance, ER rates were never below -36.4 g  $O_2$  m<sup>-2</sup> d<sup>-1</sup>, and the maximum  $\Delta$ ER was around 100% indicating that storm-driven increases in ER never exceeded twice the prior baseflow rates. These observed constraints on the heterotrophic response to storm disturbances should be interpreted as empirical bounds within our dataset rather than definitive evidence of metabolic saturation in Fuirosos

Our results also show that metabolic rates have high resilience to hydrological perturbations, as the average recovery time (RT) for GPP and ER after a storm was ca. 3 days. Similar metabolic recovery times (ca. 5 days) were observed in other USA and European headwater streams (e.g., Roberts et al., 2007; O'Connor et al., 2012; Griffiths et al., 2013). For GPP, resilience was not influenced by any hydrological descriptor, suggesting that recovery patterns might be related to other environmental drivers. For instance, previous studies have suggested that the recovery time of GPP varies with seasonal patterns of light availability (Uehlinger and Naegeli, 1998; Connell, 1978; Acuña et al., 2004; O'Donnell and Hotchkiss, 2022). For ER, recovery times increased with the storm magnitude ( $\Delta Q$ ) and the degree of stimulation ( $\Delta ER$ ), emphasizing a link between the resistance and resilience of stream biota to hydrological perturbations. Although we could not establish a clear threshold in the recovery time of ER (RT<sub>ER</sub>) with increasing the magnitude of storm events (ΔQ), RT<sub>ER</sub> never exceeded 6 days (Fig. 5d), a period clearly shorter than the mean duration of storm events in this stream (13  $\pm$  7 days). This result can be explained either by the short duration of the washing of nutrients and organic matter from terrestrial sources to the stream (Jowett and Biggs, 1997; Butturini et al., 2005; Demars, 2019), or the dilution of those resources by other catchment water sources contributing to stream runoff later on (Bernal et al., 2006). Further, this result suggests that, beyond a certain disturbance magnitude, the time needed for metabolic rates to return to baseline conditions stabilizes, potentially due to limits in biofilm recovery time or the temporal window of resource availability following storms. Overall, our findings suggest that metabolic rates, particularly ER, are susceptible to fluctuations in resource availability, especially in oligotrophic headwater streams.

#### **5 Conclusion**







This study reveals that the dynamic response of stream metabolism to storm disturbances is a key element for understanding ecosystem functioning in the context of global change. By assessing the effects of storms of varying magnitude, duration, and intensity, we gained valuable insights into how hydrological perturbations shape key metabolic processes in heterotrophic headwater streams. Our results highlight the low resistance and relatively high resilience of ER, which consistently showed significant increases during storm events (up to twice the prior rates) and relatively short recovery times (i.e., up to 6 days).

These findings highlight the importance of terrestrial-aquatic linkages in replenishing limiting resources in oligotrophic systems, where nutrient and organic matter inputs are crucial for sustaining metabolic activity. In this context, non-perennial oligotrophic streams experience "breathing storms", where disturbances stimulate heterotrophic metabolism and fundamentally alter the role of streams in carbon dioxide emissions and the global carbon cycle. To better understand the consequences and underlying mechanisms of these metabolic pulses, future studies should focus on analysing how the variability of water chemistry and dissolved organic matter composition influences stream metabolism during storms to better pinpoint the specific limiting resources driving these changes. Additionally, it is crucial to consider the stoichiometric balance of nutrients and organic matter during floods, as this balance governs the metabolic response of heterotrophic systems. Overall, our findings have significant implications for the carbon budgets of headwater catchments, as storm-driven stimulation of ER represents rapid pulses of organic carbon mineralization and emissions to the atmosphere. The magnitude and frequency of these "breathing storms" could influence not only annual carbon fluxes but also their temporal dynamics, with potential downstream effects on hydrological carbon transport. In this sense, shifts in hydrological regimes due to climate change could modify the timing, intensity, and cumulative magnitude of storm-induced metabolic pulses, thereby altering key pathways of aquatic carbon cycling.

# Data availability






The data that support the findings of this study will be openly available in a public repository (HydroShare) that issues datasets with DOIs if the paper is accepted.

# **Author contributions**

Conceptualization: CJ, AL, SB. Field work and laboratory analysis: CJ, AL, XP, SB. Data analysis: CJ, EL, JLJL, GRR. Preparation of figures and tables: CJ. Data interpretation: CJ, AL, EL, XP, JLJL, GRR, SB. Writing first draft: CJ, AL, SB. Writing final draft: CJ, AL, EL, XP, JLJL, GRR, SB.

#### 430 Competing interests

The authors declare that this research was conducted in the absence of any commercial or financial relationships that could be construed as potential conflicts of interest.

#### Acknowledgements

CJ work was supported by an FPU grant funded by the Spanish Ministry of Science, Innovation, and Universities (MICIU) (FPU21/03523). SB and XP work was supported by CSIC, MICIU, Agencia Española de Investigación (AEI), and the

European Community (FEDER UE and Next Generation fundings) through the projects EVASIONA (PID2021-122817-NB-100), RIPAMED (CNS2023-144737), BREATHE-Water4All (PCI2025-163221 /MICIU/AEI/10.13039/501100011033), and 2024ICT235. AL work was supported by the projects INHOT (CNS2022-135690) and FLUPRINT (EUR2023-143456), MICIU/AEI/FEDER UE and Next Generation funding. SB and AL also received funding from OXIT (PID2024-158111NB-C21), MICIU/FEDER UE. JLJL was supported by a Ramón y Cajal grant (RYC2022-035220-1) funded by MCIN/AEI/10.13039/501100011033 and FSE+. GRR was supported by a grant from the Swedish research Council (#2021-06667) and by the European Union (ERC, ARIMETH, 101161308). We thank Montserrat Soler for assistance with data harmonization at CEAB-CSIC and Alba Camacho for sharing data from her propane additions. Views and opinions expressed are however those of the authors only and do not necessarily reflect those of the European Union or the European Research Council. Neither the European Union nor the granting authority can be held responsible for them.

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
