# Peer review of "Breathing Storms: Enhanced Ecosystem Respiration During Storms in a Heterotrophic Headwater Stream"

_EGUsphere, 2025_

## Author Comment (AC1)

Dear editor and reviewers,

Thank you for your time and for the constructive feedback on our manuscript titled *"Breathing Storms: Enhanced Ecosystem Respiration During Storms in a Heterotrophic Headwater Stream"*. We appreciate your thoughtful and insightful comments, which will help us improve the quality and clarity of the manuscript. Below, we respond to each comment and provide an explanation of our plans to address them. The original text of the decision letter is in black and italics, while our responses are in dark blue.

**From RC1**

*Jativa et al. present an elegant study on stream metabolic rates during storm events from continuous data collection in a non-perennial Mediterranean stream. The framing of the story is logical and methods clearly address the narrative throughout the manuscript. Indeed, this contributes to a small but intriguing literature on resistance and resilience of ecosystem function in rivers. I have no large comments but raise a handful of additions to improve the clarity of the methods in the specific comments below and a comment on addressing temporal variability in metabolic patterns in rivers that could be expanded on in the introduction.*

Many thanks for your positive and encouraging comments. We are glad that you find our study elegant and clear, and we will work carefully to incorporate your comments in the revised version of the manuscript.

*L30: 'regulates'*

Thanks for noticing.

*L50: 'triggers' and 'stoppers' seem like unnecessary potential jargon. Is there another schema or metaphor that could be used?*

We appreciate your concern about potential jargon. However, and considering that R2 was prone to these concepts, and actually asked to better frame and emphasize these ideas, we have chosen to keep the terms *"metabolic triggers"* and *"metabolic stoppers"* in the revised version of the manuscript. We believe that these are easy-to-catch concepts that can be used by other authors in future studies to describe the contrasting effects of storm events on stream metabolism. To ensure that these terms are interpreted as metaphorical and not as technical jargon, we will add a clarification sentence that makes the rationale behind the terminology clearer.

*L71: I have no disagreement with any of the introduction to this point, but I think the strong temporal variability in GPP and ER need to be emphasized as potential variability to deal with in identifying resistance or resilience. A wide range of recent literature have shown within year and across year variability in GPP that are influenced by river size, hydrologic variability, and light availability (e.g., Savoy et al. 2019; Marzolf et al. 2024). I would also*

*recommend citing Lowman et al. 2024 as an example of identifying recovery of GPP in response to storm events across large scales.*

Thanks for the suggested readings; these three papers are very interesting and relevant for our study. We agree that temporal variability in stream metabolic rates could influence the detection and interpretation of resistance and resilience to storm events. However, note that our estimates of resistance and resilience are expressed as relative changes of GPP and ER to pre-event metabolic rates. Thus, any potential temporal variability in metabolic rates did not influence our estimates. Following your recommendation, we will carefully consider the best place to cite these previous studies and include additional text if needed, either in the introduction or the discussion, to better highlight the natural intrannual and interannual variability of metabolic rates and contextualize our findings.

*L116: reviewer preference for 'concentration' instead of 'levels'*

Thanks for noticing.

*L122: odd wording. Maybe change to 'we installed a monitoring station in the stream with upstream area of 9.9 km2'.*

Thanks for noticing.

*L125: what is average depth in this case? In a stilling well or staff gauge? Or is this hydraulic depth of the 200 m upstream reach? Are the pools located in areas that may alter or disrupt advective flow and create longitudinal heterogeneity in DO patterns (Rexroade et al. 2025)?*

We agree with the reviewer that stream depth might vary along the 200-m reach. In this study, average stream depth was estimated from water level measurements recorded by a pressure sensor installed in a stilling well. To verify the representativeness of this value, we conducted manual depth measurements at random transects across the reach every two weeks. We will clarify this procedure in the methods section.

We also agree that pools can generate longitudinal heterogeneity in stream flow, and that this could influence stream metabolic measurements, especially during low flow periods. To assess this effect, we installed multiple DO sensors along the reach during the transition from wet to dry conditions in 2023. These data showed similar DO patterns at the top and at the bottom of the reach, suggesting that the pools did not disrupt advective flow at the scale of our metabolism measurements during that particular period. Given that our analyses focus on storm events, when longitudinal connectivity is likely higher than during the transition from wet-to dry conditions, we are certain that the influence of slow-flow zones on DO dynamics was negligible in the present study.

*L129: how was lux converted to PPFD? This is an increasingly common practice in the literature and readers would benefit from specifics on how this was done for use in their own studies.*

In the revised Methods section, we will clarify that lux values were converted to photosynthetic photon flux density (PPFD, $\mu$mol m$^{-2}$s$^{-1}$) using a conversion factor of 0.0185, which represents the approximate conversion in forested areas such as ours.

*L150: What value of Q during the storm event was used in calculating RC? Or is it the total water flux during the storm (i.e., the integral of stream flow/total precipitation)? A few more details would be welcome as this is a potentially useful metric for others to use.*

We will include in the text that, for calculating RC, we used the total water flux during each storm event, estimated as the integral of discharge (Q) over the storm duration (total precipitation). This approach captures the cumulative effect of the storm on streamflow, rather than relying on a single value such as peak or mean discharge.

*L155: this is a great presentation of metabolism data collection and modeling. One addition I would like to see is how mean depth was determined. Mean depth is the average cross-sectional depth of the upstream contributing reach, as is defined in this study as the 200 m upstream of their sensor installation. Mean depth is often the most difficult measure to obtain from a stream reach and across flow conditions but can be estimated in similar ways with rating curves and presumably available with the data collected for the propane injections. I would like to see 1-2 sentences added to this section describing how mean depth was determined. And another sentence on QAQC approaches to the continuous data and how DO.sat was calculated too (basically address how each of the inputs to streamMetabolizer were prepared).*

Thanks for the positive comment. In the revised manuscript, we will add a brief explanation of how we calculated all inputs for *streamMetabolizer* (light, water temperature, depth, DO, and DO saturation). In particular, depth was estimated by measuring pressure in the water column every 10 minutes using a HOBO water level logger and correcting these data for atmospheric pressure using a paired barometric logger. We then calibrated the water level data with manual depth measurements taken biweekly at the study reach throughout the whole sampling period.

We will also clarify how dissolved oxygen saturation (DO.sat, in mg L$^{-1}$) was calculated. To do so, we used the standard solubility function from García and Gordon (1992), which estimates DO.sat from 10-minute data on water temperature (°C) and barometric pressure (mm Hg).

Finally, we will include a sentence describing our QA/QC procedure for all high-frequency sensor data used as input to the metabolism model. Briefly, we conducted preliminary data cleaning by removing clearly erroneous values (e.g., negative DO concentrations or spikes

from sensor fouling) and then removing noise and outliers using the *loess* R package. Specifically, we applied a locally estimated scatterplot smoothing (LOESS) model with a span parameter of 0.03 to DO, water temperature, light intensity, water depth, and DO.sat variables, effectively smoothing fluctuations and replacing outliers. The detailed step-by-step QA/QC process will be described in the Supplementary Materials.

*L166: this is a great way of constraining K in the model inputs and a great example for future researchers to approach single-site evaluations. How well does the coverage of propane injections cover the hydroperiod in the stream? These injections are often biased towards lower flows for logistical reasons, but I wonder how well empirical measures were obtained at higher flows? And as you say in L174, getting metabolism estimates during highest flows is difficult or impossible based on data and/or the models failing to converge on days with high flows.*

We were able to conduct propane additions across a wide range of flows, from 0.6 to 32 L $s^{-1}$. Logistic constraints would have been an issue, but the truth is that during the period in which we were able to conduct gas additions (2022-2024), there was an intense drought that precluded us from performing propane additions at higher discharges. We don't discard the idea of conducting more propane additions in the future to better constrain this parameter. Nevertheless, and for the sake of this study, note that we verified the accuracy of the empirically measured $K_{600}$ values by comparing them with independent estimates obtained from both the night-time regression method (Odum, 1956) and hydraulic geometry-based predictions (Raymond et al., 2012). This exercise showed that $K_{600}$ obtained from propane additions were similar to those estimated with the hydraulic geometry method (Raymond et al., 2012), and thus, we are confident about the robustness of our $K_{600}$-Q relationship. To further clarify this procedure, we will include the comparison curves showing $K_{600}$ values from propane additions alongside those predicted by the other two methods in the Supplementary Materials.

*L206: subscript PImax as is in L194*

Thanks for noticing.

*L280: might nit-pick on the 'biota' part of the response. Yes, organisms from bacteria to macro-fauna contribute to ecosystem metabolism, particularly ER, but with that statement, I would anticipate some measure of re-colonization of organisms post-storm events, whereas the response variable in this study is integrative ecosystem-scale metabolic function.*

We acknowledge that our study evaluates stream metabolism as an integrative, ecosystem-scale function, rather than tracking biotic recovery directly. To better reflect the nature of our response variable, we will revise this sentence and refer to **stream functional processes** rather than biota, thus avoiding any potential misinterpretation.

*Figure 1) should the caption for the orange dot also include 'ER'?*

We included only GPP for the orange dot because it represents a light-limited scenario, which affects GPP but not ER. Since ER is not directly influenced by light availability, it was only associated with the blue line.

*Figure 4) It maybe my computer screen but it's difficult to see the non-filled circles against the filled circles. Might recommend a different, contrasting color. Also, purely aesthetic, but can the x-axis be extended to 1000? An additional component that may help the reader discern the relationship with flow: could a vertical line be added where the typical storm flow begins? Or where is the typical baseflow? This would create a part of the graph with baseflow or losing flow metabolism could be easily compared with gaining or stormflow metabolism. If there is not a single or narrow range of flows that separate base from storm flows, disregard this final comment.*

Thanks for your suggestions. We will extend the x-axis to 1000 to enhance visual clarity, and we will remove the points corresponding to estimates that failed quality checks, as they may be misleading.

Regarding the suggestion to include a vertical line to separate baseflow from stormflow conditions, we agree that such a reference could be helpful. However, given the wide range of storm magnitudes observed in our dataset, there is no consistent discharge threshold that marks the onset of stormflow. Instead, we will highlight the days prior to each storm (i.e., baseflow conditions) using a different color in the figure. This change will allow for clearer visual distinction between baseflow and stormflow conditions while accounting for the variability in stream discharge across events.

*Figure 5) Just to be sure, the lines of best fit are coming from the methods text L193-199? What model comparison or evaluation was done to determine linear, logarithmic, or exponential was the 'best' fit to the data? Could all the evaluations be compiled into a supplementary table, perhaps with AIC and AICw values?*

The lines of best fit shown in Figure 5 correspond to the model types described in lines 193–199, selected based on the best fit to the data. Following your recommendation, we will include a supplementary table summarizing the model comparisons to increase transparency.

*From RC2*

*General comments: The authors investigate how storm events influence stream metabolism, GPP and ER, in a headwater stream, by using high-frequency DO, hydrological, and environmental measurements to analyze 35 storm events, applying Bayesian modeling. A key strength of this study lies in its robust, high-resolution dataset, which allows for a detailed examination of metabolic dynamics. The clear finding is that most analyzed storms (those with Q < 100 L/s) act as "metabolic triggers" significantly stimulating ER and demonstrating a positive relationship between ER stimulation (ΔER) and storm magnitude (ΔQ). The second finding is also very nice information about the quantification of metabolic resilience, particularly the finding that ER recovery time increases with storm magnitude but appears to saturate around 6 days.*

*Despite these strengths, the manuscript requires major revisions to address the conceptual framework established in the Introduction fully and to enhance the robustness and transparency of its interpretations. Specifically, revisions should focus on (1) evaluating the concept of metabolic saturation introduced in the Introduction section, (2) addressing the implications of excluding high-flow data (Q>100 L/s) for testing the "stopper" hypothesis and the overall representativeness of the findings, and (3-optional) acknowledging uncertainty related to gas exchange estimation during dynamic conditions.*

Many thanks for your positive and insightful comments. We are glad that you find our dataset robust and our findings nice and clear. We will work to improve the revised version of the manuscript following your suggestions and comments. Please, find below our responses to your specific queries regarding how to improve the discussion of the metabolic saturation concept and the implications of excluding the high-flow data.

*Specific comments:*

*Lines 58: introduces an interesting question about River Network Saturation concept. However, the Results section, the authors only focus on the positive linear relationship found between ΔER and ΔQ, and the Discussion does not revisit whether the data showed signs of approaching or reaching this saturation/asymptote.*

*Was the ecosystem's processing capacity likely exceeded in the largest analyzed storms, or was the range insufficient to observe this? The authors may explore more Figure 5b, such as whether the observed range of storm magnitudes was likely sufficient or insufficient to induce metabolic saturation in this system. It seems that in Figure 5d, there is a visual saturation, but this is not the concept the authors introduced in the Introduction. Please clearly differentiate the observed saturation in recovery time from the lack of observed saturation in the magnitude of the ER response.*

We appreciate this insightful comment. In the revised discussion, we will clearly state that we did not observe saturation in ER stimulation (ΔER) across the range of storm discharges

analyzed. The River Network Saturation concept has been tested and empirically proved in other fluvial systems, mostly using in-stream nutrient processing rates (Wollheim et al, 2018). Following your suggestions, we will re-analyze the data to determine if we did not observe saturation because (a) discharge never acts as a "stopper" in our system or (b) we could not estimate metabolic rates at high flows. For instance, we plan to examine the relationship between Q and ΔER using asymptotic models or breakpoint analyses to assess if there really is a lack of evidence for saturation across the observed range. Finally, we will add a discussion regarding the River Network Saturation in the new manuscript.

Moreover, we agree with the reviewer that the saturation-like response observed in ER recovery time is a separate phenomenon, not related to the expectations derived from the River Network Saturation hypothesis. We will make this point clear in the revised version of the manuscript. Specifically, we will emphasize that this pattern likely reflects a threshold in metabolic resilience, whereby the system returns to pre-storm conditions within approximately one week, regardless of further increases in storm magnitude. This concept is introduced in line 83 of the introduction, as we anticipated that recovery time may reach a threshold corresponding to the time required for biofilms to rebuild after large storm disturbances or the extended influence of nutrient and organic matter inputs, We will clarify the difference between these two phenomena throughout the revised version of the manuscript.

*Line 170: All the estimates with Q>100L/s  were excluded due to failed QC checks. I agree that the exclusion of high-flow data (>100 L/s) is based on the reported QC failures, but I am not sure if this action may prevent an empirical test of the "stopper" hypothesis. In the Introduction, lines 60-65, "Finally, during large storm events, [...] decreasing mean water residence time, scouring the benthic biomass, [...] reduce in-stream processing". These sentences refer to the "stopper" for the  large storm events, but most valid estimates were skipped to check it. Therefore, the inability to assess larger events means the full spectrum proposed in Figure 1 cannot be validated. Here are some suggestions that only use the current dataset:*

We thank the reviewer for raising this important issue and for the useful suggestions to address this point. We fully agree that the exclusion of high-flow data due to QC failures limits our ability to directly test the "stopper" end of the conceptual framework introduced in Figure 1. In order to improve this part of the discussion, we will take steps to explore how prevalent these extreme discharges are and whether useful information can still be extracted from the high-discharge events. Specifically, we are revisiting these events using adjusted model constraints and providing full access to model outputs, regardless of the QC status. With this, we aim to shed light on the potential "stopper" behavior, even if not all data at high discharges meet the standard quality thresholds required by the Bayesian model. The following responses describe the specific actions we will take to address each of the reviewer's suggestions.

1) *Report the frequency/duration of flows > 100 L/s to know the unanalyzed portion.*

In the revised Results section, we will include a summary of the occurrence of high-flow conditions. Specifically, we will report that, only 26 out of the 567 storm days analyzed exceeded 100 L s$^{-1}$ (4.6%). These days corresponded to 8 individual storm events out of the 53 analyzed (15.1%). We agree that these numbers will help contextualize the proportion of storms that could not be analyzed due to model limitations in our study stream.

2) *Table S2 does not explicitly link these failures to discharge levels. Please report more details on the QC-Failed Outputs in Supplementary Information to know which QC criteria failed.*

In the revised Supplementary Information, we will include a new table that links the range of discharge values with the specific QC criteria that were not met. This table will summarize which quality checks of the seven considered failed across discharge intervals. We agree with the reviewer that this additional information will help to clarify why metabolism estimates could not be obtained for certain high-flow events and the model limitations under dynamic conditions.

*Table R1. Summary of model performance diagnostics, showing the number of days affected by each evaluation criterion across different discharge ranges. The table reports the total number of available days, instances of unsuccessful model convergence ($n_{eff} < 8000$ or $\hat{R} > 1.2$), poor model fit ($R^2 < 0.5$ or RMSE > 0.4), and biologically implausible estimates (e.g., negative GPP or positive ER). The final column indicates the number of days that failed the quality criteria.*

| Q (L/s) | # of days | $n\_eff > 8000$ | $\hat{R}$-hat > 1.2 | $R^2 < 0.50$ | RMSE > 0.4 | GPP < 0, ER > 0 | K600 > 110 | Total of failed days |
|---|---|---|---|---|---|---|---|---|
| 0.7 - 10 | 347 | 0 | 18 | 56 | 30 | 8 | 0 | 73 |
| 10.1-40 | 135 | 0 | 10 | 18 | 23 | 11 | 0 | 29 |
| 40.1-100 | 57 | 0 | 0 | 7 | 7 | 9 | 0 | 20 |
| >100 | 26 | 0 | 0 | 19 | 19 | 14 | 18 | 26 |
| | | | | | | | Total Failed days | 148 (26%) |

3) *Figure S1 about Q-K600 relationship is very informative, but the highest discharge measured during these injections appears to be only around 32 L/s. Applying the derived Q-K600 relationship via extrapolation beyond the measured range (~32 L/s) during dynamic storm flows (up to 100 L/s) introduces uncertainty. Is it a reason for the model failing at high discharge? I recommend the SI provide a discussion about why the model likely failed QC at high flows in this system while contrasting with successful high-flow modeling in larger systems (e.g., Diamond et al., 2025a, 2025b)*

Thank you for this observation. As we addressed in our responses to RC1 (Comment L166), we acknowledge that we were unable to conduct propane additions at discharges greater than 32 L s⁻¹ due to prolonged drought during the study period. This limitation introduces some uncertainty when extrapolating the Q-$K_{600}$ relationship to high-flow conditions. In that response, we explained how we compared the propane-based $K_{600}$ estimates with independent values from the night-time regression method and the hydraulic geometry approach to assess the robustness of the derived relationship. Please refer to that comment for a detailed explanation.

To further clarify why model performance deteriorated at high flows, Table R1 shows that most days failing QC did so because the model produced unrealistic estimates (e.g., negative GPP or positive ER) or because modeled DO patterns diverged substantially from observed diel dynamics. These issues likely stem from sensor displacement or burial during turbulent flow, or from diel variability in metabolic (GPP, ER) or physical ($K_{600}$) parameters not captured by the model.

It is also important to note that Fuirosos is a small, shallow headwater stream with a median discharge of 12 L s⁻¹ and a median depth of 7.5 cm. Flow events exceeding 100 L s⁻¹ are rare and represent an order-of-magnitude increase over baseflow, often resulting in overbank flooding and complex hydrodynamics that can disrupt DO signals and violate model assumptions. In contrast, studies such as Diamond et al. (2025) involve larger, deeper systems where similar flow increases produce less drastic hydromorphological changes.

As recommended, we will include a brief discussion of these points, along with the comparison plots, in the Supplementary Information.

*4)    I would like to see output distributions (credible intervals/ranges) for GPP/ER/K600 for all these high-flow runs (Q>100 L/s). Even though the median values failed for QC, using the credible intervals may give us some helpful information, such as the system is more "stoppers" or more "triggers" behavior at these high flows.*

*--> I suggest that authors may explicitly state the "stopper" hypothesis remains empirically untested by reliable data from this study for higher flows if using credible intervals output still does not give any further information.*

We thank the reviewer for this constructive suggestion. We will include in a public repository (HydroShare) the full set of metabolism model outputs from streamMetabolizer, including all GPP, ER, and $K_{600}$ estimates, regardless of whether they passed the quality check. This will allow readers to evaluate the full distribution of outputs, including those from high-flow events (>100 L s⁻¹), and to interpret model behavior beyond the subset of accepted values.

Following the reviewer's recommendation, we will also re-run the high-flow events (n = 26 days in total) using adjusted model constraints, specifically by setting feasible upper limits for depth and $K_{600}$ based on our empirical data. This approach may allow us to "rescue" some estimates that were previously rejected due to the uncertainty associated with these two parameters. From preliminary trials, we have found that for most high-flow days where the quality checks failed, model outputs remained unreliable even after doing some adjustments for depth and $K_{600}$ (see example figure below). However, in a subset of cases where the failure was due to $K_{600}$ values exceeding 110 $d^{-1}$, these adjustments has allowed the model to converge with credible estimates. In the revised discussion, we plan to include this exercise to shed some additional light on our saturation hypothesis.

**Fig R1**. *Temporal patterns of dissolved oxygen dynamics during a period including three consecutive high-flow days (Q > 100 L/s). The panels show (top) dissolved oxygen concentration (DO, mg/L), (middle) DO saturation (%), and (bottom) the rate of change in DO (dDO/dt, mg/L/d). Bold lines represent observed values, while lighter lines indicate model predictions generated using StreamMetabolizer. The figure illustrates both the general agreement and discrepancies between observed and modeled values under high-discharge conditions.*

[Figure]

*Line 174: "did not passed" -> did not pass*
Thanks for noticing.

*Line 155 and 175: Add a brief example of ΔER calculation with negative ER. This is quite confusing when saying to increase or decrease ER while the ER is negative.*

We will add a clarifying sentence (and a simple example) in the Methods section to explain that ER values are negative by convention, and that an increase in ER (i.e., more negative) indicates a stimulation of respiration.

*Line 270-275: Consistent adding ΔMET/ΔGPP/ΔER definition. Better clarifying axis labels in Figs 5 & 6.*

We will ensure consistent use and clear definitions of ΔGPP, and ΔER as the changes in gross primary production and ecosystem respiration during storm events relative to baseflow conditions in the main text and the caption of both figures 5 and 6.

*Line 55/88: Clarify "recovery time" vs. River Network Saturation.*

Thanks, this distinction is important. In the revised manuscript, we will make clear that the River Network Saturation concept relates to the potential limit of the system to process materials during high flows—essentially a matter of how much metabolism changes can occur in response to disturbance. In contrast, the recovery time refers to how long it takes for the system to return to baseline metabolic conditions after a storm, which is related to the resilience of the system. In both cases, we can observe asymptotic behaviour but the mechanisms are different in each case.

*Line 352: "ER recovery times at ca. 6 days (Fig. 5a)" -> it should be Fig. 5d*

Thanks for noticing.

**References**

Diamond, J. S., Bernal, S., Boukra, A., Cohen, M. J., Lewis, D., Masson, M., Moatar, F., & Pinay, G.:Stream network variation in dissolved oxygen: Metabolism proxies and biogeochemical controls, *Ecological Indicators*, 131, 108233. https://doi.org/10.1016/j.ecolind.2021.108233, 2025.

Raymond, P. A., Zappa, C. J., Butman, D., Bott, T. L., Potter, J., Mulholland, P., Laursen, A. E., McDowell, W. H., and Newbold, D.: Scaling the gas transfer velocity and hydraulic geometry in streams and small rivers, *Limnology and Oceanography: Fluids and Environments,* 2, 41–53, https://doi.org/10.1215/21573689-1597669, 2012.

Odum, H. T.: Primary Production in Flowing Waters, *Limnology and Oceanography,* 1, 102–117, https://doi.org/10.4319/lo.1956.1.2.0102, 1956.

Wollheim, W. M., Bernal, S., Burns, D. A., Czuba, J. A., Driscoll, C. T., Hansen, A. T., Hensley, R. T., Hosen, J. D., Inamdar, S., Kaushal, S. S., Koenig, L. E., Lu, Y. H., Marzadri, A., Raymond, P. A., Scott, D., Stewart, R. J., Vidon, P. G., and Wohl, E.: River network saturation concept: factors influencing the balance of biogeochemical supply and demand of river networks, *Biogeochemistry,* 141, 503–521, https://doi.org/10.1007/s10533-018-0488-0, 2018.

---

## Author Response (AR1)

Dear editor and reviewers,

Thank you for your time and for the constructive feedback on our manuscript titled "Breathing Storms: Enhanced Ecosystem Respiration During Storms in a Heterotrophic Headwater Stream". We appreciate your thoughtful and insightful comments, which will help us improve the quality and clarity of the manuscript. Below, we respond to each comment. The original text of the decision letter is in black and italics, while our responses are in dark blue.

**From RC1**

Jativa et al. present an elegant study on stream metabolic rates during storm events from continuous data collection in a non-perennial Mediterranean stream. The framing of the story is logical and methods clearly address the narrative throughout the manuscript. Indeed, this contributes to a small but intriguing literature on resistance and resilience of ecosystem function in rivers. I have no large comments but raise a handful of additions to improve the clarity of the methods in the specific comments below and a comment on addressing temporal variability in metabolic patterns in rivers that could be expanded on in the introduction.

Many thanks for your positive and encouraging comments. We are glad that you find our study elegant and clear, and we have worked carefully to incorporate your comments in the revised version of the manuscript.

L30: 'regulates'

We have changed to "regulates" as suggested.

L50: 'triggers' and 'stoppers' seem like unnecessary potential jargon. Is there another schema or metaphor that could be used?

We appreciate your concern about potential jargon. However, and considering that Reviewer 2 was prone to these concepts, and actually asked to better frame and emphasize these ideas, we have chosen to keep the terms "metabolic triggers" and "metabolic stoppers" in the revised version of the manuscript. We believe that these are easy-to-catch concepts that can be used by other authors in future studies to describe the contrasting effects of storm events on stream metabolism. To ensure that these terms are interpreted correctly, we have added some clarification in the introduction to make the rationale behind clearer. Updated text (L52-56):"... increasing discharge can act either as a "metabolic trigger" or as a "metabolic stopper" depending on the balance between biological processes and physical disruptive forces. A storm event can trigger stream metabolic rates by supplying limiting resources, such as nutrients and organic matter. Conversely, a storm event can inhibit metabolic activity either through biofilm scouring or by increasing turbidity and reducing light availability."

L71: I have no disagreement with any of the introduction to this point, but I think the strong temporal variability in GPP and ER need to be emphasized as potential variability to deal with in identifying resistance or resilience. A wide range of recent literature have shown within year and across year variability in GPP that are influenced by river size, hydrologic variability, and light availability (e.g., Savoy et al. 2019; Marzolf et al. 2024). I would also recommend citing Lowman et al. 2024 as an example of identifying recovery of GPP in response to storm events across large scales.

Thanks for the suggested readings; these three papers are very interesting and relevant for our study. Following your recommendation, we have added these citations to the revised manuscript. Further, we agree that temporal variability in stream metabolic rates could influence the detection and interpretation of resistance and resilience to storm events. However, note that our estimates of resistance and resilience are expressed as relative changes of GPP and ER to pre-event metabolic rates. Thus, any potential temporal variability in metabolic rates did not influence our estimates.

L116: reviewer preference for 'concentration' instead of 'levels'

We have changed to "concentration" as suggested

L122: odd wording. Maybe change to 'we installed a monitoring station in the stream with upstream area of 9.9 km2'.

We have changed the sentence to "... we installed a monitoring station in the stream with an upslope contributing area of 9.9 km²"

L125: what is average depth in this case? In a stilling well or staff gauge? Or is this hydraulic depth of the 200 m upstream reach? Are the pools located in areas that may alter or disrupt advective flow and create longitudinal heterogeneity in DO patterns (Rexroade et al. 2025)?

We agree with the reviewer that stream depth might vary along the 200-m reach. In this study, average stream depth was estimated from water level measurements recorded by a pressure sensor installed in a stilling well. To verify the representativeness of this value, we conducted manual depth measurements at random transects across the reach every two weeks. We have clarified this procedure in the methods section. Updated text (L137-141):

"Stream water depth (h, in cm) was estimated from 10-minute pressure measurements using a HOBO water level logger installed in a stilling well, with atmospheric pressure corrections applied using a paired barometric logger located in a nearby tree. To verify the representativeness of these data across the 200-m study reach, manual depth measurements were conducted biweekly at randomly selected transects. These manual measurements were also used to calibrate the water level data, confirming that the sensor reliably captured overall depth variability during both baseflow and storm conditions."

We also agree that pools can generate longitudinal heterogeneity in advective flow and that this could influence stream metabolic measurements, especially during low flow periods. To assess this effect, we installed two DO sensors along the reach during the transition from wet to dry conditions in 2023. These data showed similar DO patterns at the top and the bottom of the reach, suggesting that the pools did not disrupt advective flow at the scale of our metabolism measurements during that particular period. Given that our analyses focus on storm events, when longitudinal connectivity is higher than during the transition from wet to dry conditions, we are certain that the influence of slow-flow zones on DO dynamics was negligible in the present study.

L129: how was lux converted to PPFD? This is an increasingly common practice in the literature and readers would benefit from specifics on how this was done for use in their own studies.

In the revised Methods section, we have clarified this point as follows (L134-135):

"Lux values from the two sensors were averaged and converted to photosynthetic photon flux density (PPFD, in  $\mu$ mol  $m^{-2}$   $s^{-1}$ ) using a conversion factor of 0.0185, which is commonly applied for forested environments (Thimijan and Heins, 1983). The resulting PPFD values were then aggregated into daily totals."

L150: What value of Q during the storm event was used in calculating RC? Or is it the total water flux during the storm (i.e., the integral of stream flow/total precipitation)? A few more details would be welcome as this is a potentially useful metric for others to use.

In the revised Methods section, we have clarified this point as follows (L160-162):

"The RC was calculated using the total water flux in mm during each storm event, estimated as the integral of Q over P. This cumulative approach accounts for antecedent moisture conditions and captures the stream's overall hydrological response to each storm event, avoiding dependence on potentially unrepresentative single-point Q values."

L155: this is a great presentation of metabolism data collection and modeling. One addition I would like to see is how mean depth was determined. Mean depth is the average cross-sectional depth of the upstream contributing reach, as is defined in this study as the 200 m upstream of their sensor installation. Mean depth is often the most difficult measure to obtain from a stream reach and across flow conditions but can be estimated in similar ways with rating curves and presumably available with the data collected for the propane injections. I would like to see 1-2 sentences added to this section describing how mean depth was determined. And another sentence on QAQC approaches to the continuous data and how DO.sat was calculated too (basically address how each of the inputs to streamMetabolizer were prepared).

Thanks for the positive comment. In a previous comment we already addressed how we measured and validated mean water depth and presented a corrected version of the text in the manuscript (L137-141). Moreover, we have added this clarification (L172-176):

"All inputs for the streamMetabolizer model (i.e., light, water temperature, depth, DO, and DO saturation) were derived from the high-frequency sensor data described in Section 2.2. DO saturation was calculated from 10-minute water temperature and barometric pressure values using the García and Gordon (1992) solubility equation. Sensor data were quality-checked by removing implausible values (e.g., negative DO, sensor spikes) and smoothed using LOESS regression (span = 0.03) to reduce noise and filter outliers."

Finally, in the Supplementary Materials (Appendix 1), there is a section called "Data preparation" detailing the step-by-step QA/QC process.

L166: this is a great way of constraining K in the model inputs and a great example for future researchers to approach single-site evaluations. How well does the coverage of propane injections cover the hydroperiod in the stream? These injections are often biased towards lower flows for logistical reasons, but I wonder how well empirical measures were obtained at higher flows? And as you say in L174, getting metabolism estimates during highest flows is difficult or impossible based on data and/or the models failing to converge on days with high flows.

We were able to conduct propane additions across a wide range of flows, from 0.6 to 32 L s-1. Logistic constraints would have been an issue, but the truth is that during the period in which we were able to conduct gas additions (2022-2024), there was an intense drought that precluded us from performing propane additions at higher discharges. We don't discard the idea of conducting more propane additions in the future to better constrain this parameter. Nevertheless, and for the sake of this study, note that we verified the accuracy of the empirically measured K600 values by comparing them with independent estimates obtained from both the night-time regression method (Odum, 1956) and hydraulic geometry-based predictions (Raymond et al., 2012). This exercise showed that K600 obtained from propane additions were similar to those estimated with the hydraulic geometry method (Raymond et al., 2012). Thus, we are confident about the robustness of our K600-Q relationship. To further clarify this procedure, we have included the comparison curves showing K600 values from propane additions alongside those predicted by the other two methods in the Supplementary Materials (Figure S1b).

Figure S1. Mean daily discharge (Q) and gas exchange coefficient ( $K_{600}$ ). (a)  $K_{600}$  estimates from 11 propane tracer additions using a mixed tracer injection method (Jin et al., 2012). A logarithmic relationship between Q and  $K_{600}$  was derived from these measurements. (b)  $K_{600}$  estimates for all days before, during, and after storm events using three methods: night-time regression (yellow dots, n = 417), hydraulic geometry following Raymond et al. (2012) (grey dots, n = 774), and modeled  $K_{600}$  values (black dots) calculated using the  $Q-K_{600}$  relationship derived from panel (a).

L206: subscript PImax as is in L194

We have changed to "PImax" as suggested

L280: might nit-pick on the 'biota' part of the response. Yes, organisms from bacteria to macro-fauna contribute to ecosystem metabolism, particularly ER, but with that statement, I would anticipate some measure of re-colonization of organisms post-storm events, whereas the response variable in this study is integrative ecosystem-scale metabolic function.

We acknowledge that our study evaluates stream metabolism as an integrative, ecosystem-scale function, rather than tracking biotic recovery directly. To better reflect the nature of our response variable, we have revised this sentence and refer to **stream processes** rather than biota, thus avoiding any potential misinterpretation (L302).

Figure 1) should the caption for the orange dot also include 'ER'?

We included only GPP for the orange dot because it represents a light-limited scenario, which affects GPP but not ER. Since ER is not directly influenced by light availability, it was only associated with the blue line.

Figure 4) It maybe my computer screen but it's difficult to see the non-filled circles against the filled circles. Might recommend a different, contrasting color. Also, purely aesthetic, but can the x-axis be extended to 1000? An additional component that may help the reader discern the relationship with flow: could a vertical line be added where the typical storm flow begins? Or where is the typical baseflow? This would create a part of the graph with baseflow or losing flow metabolism could be easily compared with gaining or stormflow metabolism. If there is not a single or narrow range of flows that separate base from storm flows, disregard this final comment.

Thanks for your suggestions. We have removed the points corresponding to estimates that failed quality checks, as they may be misleading to the reader.

Regarding the suggestion to include a vertical line to separate baseflow from stormflow conditions, we agree that such a reference could be helpful. However, given the wide range of storm magnitudes observed in our dataset, there is no consistent discharge threshold that marks the onset of stormflow. Instead, we have highlighted the days prior to each storm (i.e., baseflow conditions) using a different color in new Figure 4. This change allows for a clearer visual distinction between baseflow and stormflow conditions while accounting for the variability in stream discharge across events.

Figure 4. Relationship between mean daily discharge (Q) and daily a) gross primary production (GPP) and b) ecosystem respiration (ER) across all individual storm events (in blue) and for the days 1 to 6 prior to each event (orange). The black line represents the linear regression between variables (only shown when p < 0.01) and the shaded area indicates the 95% confidence interval. Note that regressions were based only on storm event days.

Figure 5) Just to be sure, the lines of best fit are coming from the methods text L193-199? What model comparison or evaluation was done to determine linear, logarithmic, or exponential was the 'best' fit to the data? Could all the evaluations be compiled into a supplementary table, perhaps with AIC and AICw values?

Thank you for this suggestion. We have included a supplementary table summarizing the model comparisons to enhance transparency (Table S4). After conducting AIC-based evaluations, we found that linear and logarithmic models were equally supported in two cases: the relationships between  $\Delta Q$  and  $\Delta ER$ , and between  $\Delta Q$  and  $RT_{ER}$ . These findings have been incorporated into the Results section, and Figure 5 has been updated accordingly to show both relationships. Additionally, we chose not to include exponential models, as they were not significant in any case and did not align with our expectations.

Table S4. Comparison of linear and logarithmic regression model performance comparing hydrological, environmental, and biological variables. Each row shows a pair of variables tested as predictors and responses. For each model type, the Akaike Information Criterion (AIC), p-value, and coefficient of determination ( $R^2$ ) are reported. Predictor variables include maximum rainfall intensity ( $PI_{max}$ ), storm duration (D), runoff coefficient ( $R^2$ ), change in discharge (D), daily discharge (D), temperature (D), and light ( $D^2$ ). Response variables include gross primary production ( $D^2$ ), ecosystem respiration ( $D^2$ ), metabolic resistance as the change in metabolic rates between the storms and pre-storm conditions ( $D^2$ ), and metabolic resilience as recovery time ( $D^2$ ),  $D^2$ ). In bold are the models that were finally selected based on AIC and p-values ( $D^2$ ) value ( $D^2$ ). When the difference in AIC between the two models was less than 2, we considered them equally supported. Missing values (--) indicate cases where logarithmic transformation was not possible.

|                       | s vaines ( ) in | Linear model |         |                | Logarithmic model |         |                |  |
|-----------------------|-----------------|--------------|---------|----------------|-------------------|---------|----------------|--|
| Relationship explored |                 | AIC          | p value | $\mathbb{R}^2$ | AIC               | p value | R 2 |  |
| PI max     | GPP             | 126          | 0.035   | 0.13           | 129               | 0.183   | 0.05           |  |
| PI max     | ER              | 232          | 0.015   | 0.17           | 234               | 0.034   | 0.13           |  |
| D                     | GPP             | 131          | 0.632   | 0.01           | 130               | 0.424   | 0.02           |  |
| D                     | ER              | 236          | 0.104   | 0.08           | 237               | 0.157   | 0.06           |  |
| RC                    | GPP             | 131          | 0.775   | 0.00           | 131               | 0.659   | 0.01           |  |
| RC                    | ER              | 238          | 0.386   | 0.02           | 239               | 0.650   | 0.01           |  |
| $\Delta Q$            | GPP             | 130          | 0.312   | 0.03           | 131               | 0.660   | 0.01           |  |
| $\Delta Q$            | ER              | 237          | 0.212   | 0.05           | 236               | 0.143   | 0.06           |  |
| Q                     | GPP             | 1405         | 0.025   | 0.01           | 1409              | 0.803   | 0.00           |  |
| Q                     | ER              | 2549         | < 0.001 | 0.21           | 2586              | < 0.001 | 0.12           |  |
| T                     | GPP             | 1903         | 0.06    | 0.01           | 1902              | 0.04    | 0.01           |  |
| T                     | ER              | 3554         | < 0.001 | 0.06           | 3552              | < 0.001 | 0.06           |  |
| PAR                   | GPP             | 128          | 0.097   | 0.08           | 125               | 0.017   | 0.16           |  |
| PAR                   | ER              | 239          | 0.734   | 0.00           | 239               | 0.753   | 0.00           |  |
| $PI_{max}$            | $\Delta GPP$    | 391          | 0.477   | 0.02           | 391               | 0.539   | 0.01           |  |
| PI max     | $\Delta ER$     | 341          | 0.352   | 0.03           | 342               | 0.862   | 0.00           |  |
| D                     | $\Delta GPP$    | 391          | 0.672   | 0.01           | 391               | 0.526   | 0.01           |  |
| D                     | $\Delta ER$     | 342          | 0.936   | 0.00           | 342               | 0.928   | 0.00           |  |
| RC                    | $\Delta GPP$    | 391          | 0.663   | 0.01           | 391               | 0.739   | 0.00           |  |
| RC                    | $\Delta ER$     | 340          | 0.216   | 0.05           | 339               | 0.103   | 0.08           |  |
| $\Delta Q$            | $\Delta GPP$    | 387          | 0.034   | 0.13           | 388               | 0.071   | 0.10           |  |
| ΔQ                    | ΔER             | 325          | < 0.001 | 0.37           | 325               | <0.001  | 0.39           |  |
| PI max     | $RT_{GPP}$      | 153          | 0.427   | 0.02           | 152               | 0.184   | 0.05           |  |
| $PI_{max}$            | $RT_{ER}$       | 150          | 0.864   | 0.00           | 150               | 0.563   | 0.01           |  |
| D                     | $RT_{GPP}$      | 151          | 0.110   | 0.08           | 151               | 0.161   | 0.06           |  |
| D                     | $RT_{ER}$       | 148          | 0.246   | 0.04           | 149               | 0.352   | 0.03           |  |
| RC                    | $RT_{GPP}$      | 153          | 0.380   | 0.02           | 153               | 0.930   | 0.00           |  |
| RC                    | $RT_{ER}$       | 147          | 0.132   | 0.07           | 148               | 0.221   | 0.05           |  |
| $\Delta Q$            | $RT_{GPP}$      | 153          | 0.490   | 0.01           | 153               | 0.822   | 0.00           |  |
| ΔQ                    | RTER            | 126          | <0.001  | 0.49           | 128               | < 0.001 | 0.46           |  |
| $RT_{GPP}$            | ΔGPP            | 387          | 0.046   | 0.11           |                   |         |                |  |
| RTER                  | ΔER             | 321          | < 0.001 | 0.45           |                   |         |                |  |

**From RC2**

General comments: The authors investigate how storm events influence stream metabolism, GPP and ER, in a headwater stream, by using high-frequency DO, hydrological, and environmental measurements to analyze 35 storm events, applying Bayesian modeling. A key strength of this study lies in its robust, high-resolution dataset, which allows for a detailed examination of metabolic dynamics. The clear finding is that most analyzed storms (those with  $Q < 100 \, \text{L/s}$ ) act as "metabolic triggers" significantly stimulating ER and demonstrating a positive relationship between ER stimulation ( $\Delta$ ER) and storm magnitude ( $\Delta$ Q). The second finding is also very nice information about the quantification of metabolic resilience, particularly the finding that ER recovery time increases with storm magnitude but appears to saturate around 6 days.

Despite these strengths, the manuscript requires major revisions to address the conceptual framework established in the Introduction fully and to enhance the robustness and transparency of its interpretations. Specifically, revisions should focus on (1) evaluating the concept of metabolic saturation introduced in the Introduction section, (2) addressing the implications of excluding high-flow data (Q>100 L/s) for testing the "stopper" hypothesis and the overall representativeness of the findings, and (3-optional) acknowledging uncertainty related to gas exchange estimation during dynamic conditions.

Many thanks for your positive and insightful comments. We are glad that you find our dataset robust and our findings nice and clear. We have worked to improve the revised version of the manuscript following your suggestions and comments. Please, find below our responses to your specific queries regarding how to improve the discussion of the metabolic saturation concept and the implications of excluding the high-flow data.

**Specific comments:**

Lines 58: introduces an interesting question about River Network Saturation concept. However, the Results section, the authors only focus on the positive linear relationship found between  $\Delta ER$  and  $\Delta Q$ , and the Discussion does not revisit whether the data showed signs of approaching or reaching this saturation/asymptote.

Was the ecosystem's processing capacity likely exceeded in the largest analyzed storms, or was the range insufficient to observe this? The authors may explore more Figure 5b, such as whether the observed range of storm magnitudes was likely sufficient or insufficient to induce metabolic saturation in this system. It seems that in Figure 5d, there is a visual saturation, but this is not the concept the authors introduced in the Introduction. Please clearly differentiate the observed saturation in recovery time from the lack of observed saturation in the magnitude of the ER response.

We appreciate this insightful comment. Following your suggestion as well as suggestions from R1, we re-analyzed the data considering AIC criteria. We found that linear and

logarithmic models were equally supported to understand the relationship between  $\Delta Q$  and  $\Delta ER$ . These findings have been incorporated into the Results section, and Figure 5 has been updated accordingly to show both relationships. Updated text (L370-376):

"Moreover, the relationship between  $\Delta ER$  and  $\Delta Q$  was equally well explained by both linear and logarithmic models, preventing us from identifying a clear saturation pattern in ER–unlike the saturation responses often reported for nutrient uptake with increasing discharge (sensu River Network Saturation concept; Wollheim et al., 2018). Nevertheless, we observed certain constraints in ER during storms. For instance, ER rates were never more negative than -36.4 g  $O_2$  m $^{-2}$  d $^{-1}$ , and the maximum  $\Delta ER$  was around 100%, indicating that storm-driven increases in ER never exceeded twice the prior baseflow rates. While these constraints do not confirm a saturation effect, they may reflect an upper limit of the heterotrophic response to storm disturbances in Fuirosos."

Moreover, we agree with the reviewer that the saturation-like response observed in ER recovery time (RTER) is a distinct phenomenon, unrelated to the expectations derived from the River Network Saturation hypothesis. Rather than indicating saturation of metabolic response, this pattern more likely reflects a threshold in system resilience. We have made this point clear in the revised version of the manuscript. Updated text (L389-391):

"Further, this result suggests that, beyond a certain disturbance magnitude, the time needed for metabolic rates to return to baseline conditions stabilizes, potentially due to limits in biofilm recovery time or the temporal window of resource availability following storms."

Line 170: All the estimates with Q>100L/s were excluded due to failed QC checks. I agree that the exclusion of high-flow data (>100 L/s) is based on the reported QC failures, but I am not sure if this action may prevent an empirical test of the "stopper" hypothesis. In the Introduction, lines 60-65, "Finally, during large storm events, [...] decreasing mean water residence time, scouring the benthic biomass, [...] reduce in-stream processing". These sentences refer to the "stopper" for the large storm events, but most valid estimates were skipped to check it. Therefore, the inability to assess larger events means the full spectrum proposed in Figure 1 cannot be validated. Here are some suggestions that only use the current dataset:

We thank the reviewer for raising this important issue and for the useful suggestions to address this point. We fully agree that the exclusion of high-flow data due to QC failures limits our ability to directly test the "stopper" end of the conceptual framework introduced in Figure 1. In order to improve this part of the discussion, we have taken steps to explore how prevalent these extreme discharges are and whether useful information can still be extracted from the high-discharge events. Specifically, we have revised these events using adjusted model constraints and providing full access to model outputs, regardless of the QC status. With this, we aim to shed light on the potential "stopper" behavior, even if not all data at high discharges meet the standard quality thresholds required by the Bayesian model. The

following responses describe the specific actions we have taken to address each of the reviewer's suggestions.

**1) Report the frequency/duration of flows > 100 L/s to know the unanalyzed portion.**

In our study, the frequency of high-flow days (>100 L s-1) represented only 26 out of the 567 storm-related days (4.6%). This limited representation constrained our ability to directly evaluate potential "metabolic stopper" effects. In the revised manuscript we have addressed this as follows Updated text (L368-370):

"During the study period, these high-flow days represented less than 5% of all the storm days analyzed and occurred during only 8 of the 53 identified storm events. Thus, large or extreme storms that might inhibit ER were relatively rare in this stream, which more commonly exhibited ER stimulation during storms."

**2) Table S2 does not explicitly link these failures to discharge levels. Please report more details on the QC-Failed Outputs in Supplementary Information to know which QC criteria failed.**

We agree with the reviewer that this additional information will help to clarify why metabolism estimates could not be obtained for certain high-flow events and the model limitations under dynamic conditions. In the revised Supplementary Information, we have updated Table S3 with the range of discharge values with the specific QC criteria that were not met. Table S3 shows that all metabolism estimates for flows >100 L/s failed to meet one or more of the quality control criteria. Specifically, model failure at these high flows was driven by both poor model fit (e.g., 73% of high-flow days had  $R^2 < 0.5$ , and 73% had RMSE > 0.4) and biologically implausible results (e.g., 54% of days with GPP < 0 or ER > 0, and 69% with unrealistically high K600 values).

Table S3. Diagnostics assessing model performance, detailing the total days analyzed (including storm days and days prior to each storm event) and storm events affected by each criterion. The table includes the total available data, the number of imputed days using miceRanger, and occurrences of biologically implausible values (i.e., negative GPP or positive ER). It also reports instances of unsuccessful model convergence ( $\hat{R}$ -hat > 1.2 and  $n_{\rm eff}$  < 8000) and days with poor model fit ( $R^2$  < 0.5, RMSE > 0.4). Finally, the number of days that passed all quality checks is indicated. The same diagnostics are shown for different ranges of discharge during storm events.

| Quality test         | Days
analyzed | Number of storm    | Discharge Range During Storm events (L/s) |                     |                   |                      |  |
|----------------------|------------------|--------------------|-------------------------------------------|---------------------|-------------------|----------------------|--|
|                      | anaryzeu         | events             | 0.7–10                                    | 10.1–40             | 40.1–100          | >100                 |  |
| Total data           | 698              | 53
6
18
8 | 347
32
8
0                       | 135
0
11
0 | 57
2
9
0 | 26
14
14
18 |  |
| Imputed data         | 48
53
18   |                    |                                           |                     |                   |                      |  |
| GPP < 0, ER > 0      |                  |                    |                                           |                     |                   |                      |  |
| $K_{600} > 110$      |                  |                    |                                           |                     |                   |                      |  |
| $\hat{R}$ -hat > 1.2 | 48               | 9                  | 18                                        | 10                  | 0                 | 0                    |  |
| $n_eff > 8000$       | 0                | 0                  | 0                                         | 0                   | 0                 | 0                    |  |
| $R^2 < 0.50$         | 147              | 31                 | 56                                        | 18                  | 7                 | 19                   |  |
| RMSE > 0.4 | 92               | 26                 | 30                                        | 23                  | 7                 | 19                   |  |
| Passed quality check | 542              | 35                 | 274                                       | 106                 | 37                | 0                    |  |
| % Success            | 72%              | 66%                | 79%                                       | 79%                 | 65%               | 0%                   |  |

3) Figure S1 about Q-K600 relationship is very informative, but the highest discharge measured during these injections appears to be only around 32 L/s. Applying the derived Q-K600 relationship via extrapolation beyond the measured range (~32 L/s) during dynamic storm flows (up to 100 L/s) introduces uncertainty. Is it a reason for the model failing at high discharge? I recommend the SI provide a discussion about why the model likely failed QC at high flows in this system while contrasting with successful high-flow modeling in larger systems (e.g., Diamond et al., 2025a, 2025b)

Thank you for this observation. As we addressed in our responses to R1, we acknowledge that we were unable to conduct propane additions at discharges greater than  $32 \text{ L s}^{-1}$  due to prolonged drought during the study period. This limitation introduces some uncertainty when extrapolating the Q-K600 relationship to high-flow conditions. In that response, we explained how we compared the propane-based K600 estimates with independent values from the night-time regression method and the hydraulic geometry approach to assess the robustness of the derived relationship. Please refer to that comment for a detailed explanation.

To further clarify why model performance deteriorated at high flows, Table S3 shows that most days failing QC did so because the model produced unrealistic estimates (e.g., negative GPP or positive ER) or because modeled DO patterns diverged substantially from observed diel dynamics. These issues likely stem from sensor displacement or burial during turbulent flow, or from diel variability in metabolic (GPP, ER) or physical (K600) parameters not captured by the model. This point is now included in the discussion (L365-368):

"Fuirosos is a small stream (median  $Q = 12 L s^{-1}$ ; median depth = 7.5 cm), where floods exceeding  $100 L s^{-1}$  often lead to overbank flooding and displacement or burial of sensors. These complex hydrodynamics and field constraints can disrupt DO signals and complicate gas exchange inference, ultimately violating key assumptions of metabolic models. These complex hydrodynamics and field constraints can disrupt DO signals and complicate gas exchange inference, ultimately violating key assumptions of metabolic models."

- 4) I would like to see output distributions (credible intervals/ranges) for GPP/ER/K600 for all these high-flow runs (Q>100 L/s). Even though the median values failed for QC, using the credible intervals may give us some helpful information, such as the system is more "stoppers" or more "triggers" behavior at these high flows.
- --> I suggest that authors may explicitly state the "stopper" hypothesis remains empirically untested by reliable data from this study for higher flows if using credible intervals output still does not give any further information.

We thank the reviewer for this constructive suggestion. We will include in a public repository (HydroShare) the full set of metabolism model outputs from streamMetabolizer, including all daily GPP, ER, and  $K_{600}$  estimates, regardless of whether they passed the quality check. This will allow readers to evaluate the full distribution of outputs, including those from high-flow events (>100 L s-1), and to interpret model behavior beyond the subset of accepted values.

Following the reviewer's recommendation, we revisited the 26 high-flow days (Q > 100 L s-1) by re-running the metabolism model using adjusted constraints, specifically narrowing the feasible range for depth and K600 based on our empirical data. We found that for most of these high-flow days, model outputs remained unreliable even after these adjustments, typically due to persistent convergence issues or implausible parameter estimates (see example figure below). However, in 8 of the 26 days, where the original model failures were specifically due to K600 values exceeding 110 d-1, the constrained model produced estimates that met our convergence and plausibility criteria. For these days, high Q ranged from 100 to 1370 L s-1 and ER values were similarly elevated (range: 20 to 41g O2 m-2 d-1) as on days with large  $\Delta$ Q increases, providing no evidence for a metabolic stopper effect within our data.

Fig R1. Temporal patterns of dissolved oxygen dynamics during a period including three consecutive high-flow days (Q > 100 L/s). The panels show (top) dissolved oxygen concentration (DO, mg/L), (middle) DO saturation (%), and (bottom) the rate of change in DO (dDO/dt, mg/L/d). Bold lines represent observed values, while lighter lines indicate model predictions generated using StreamMetabolizer. The figure illustrates both the general agreement and discrepancies between observed and modeled values under high-discharge conditions.

Line 174: "did not passed" -> did not pass
We have changed to "did not pass" as suggested

Line 155 and 175: Add a brief example of  $\Delta ER$  calculation with negative ER. This is quite confusing when saying to increase or decrease ER while the ER is negative.

We have added a clarification in the Methods section (L296-204) to explain that for the  $\Delta$ ER calculations, we used absolute values. In this sense, an increase in ER (i.e., more negative) indicates a stimulation of respiration.

Line 270-275: Consistent adding  $\Delta MET/\Delta GPP/\Delta ER$  definition. Better clarifying axis labels in Figs 5 & 6.

We have changed the captions of both figures to make it more clear.

"Figure 5. Relationship between changes in discharge ( $\Delta Q$ , defined as the difference between peak flow and baseflow) and stream metabolic responses. Panels (a) and (b) show the change in gross primary production ( $\Delta GPP$ ) and ecosystem respiration ( $\Delta ER$ ), respectively, used as proxies for metabolic resistance. Panels (c) and (d) show recovery time

for GPP (RTGPP) and ER (RTER), used as proxies for metabolic resilience. Color coding indicates whether metabolic activity was stimulated (blue), suppressed (red), or showed no significant change (yellow) in response to the storm event. Solid and dashed lines represent the best fit models, only when statistically significant (p < 0.01) and the shaded area indicates the 95% confidence interval. For panels (b) and (d), there were no differences in the goodness of fit between the logarithmic (log) and linear (lm) models (Table S4)."

"Figure 6.Relationship between resistance and resilience of gross primary production (GPP) and (b) ecosystem respiration (ER) during storm events. Resistance is the relative change in metabolic rates compared to prior baseflow conditions (i.e.,  $\Delta$ GPP and  $\Delta$ ER). Resilience is the recovery time of metabolic rates to prior conditions (i.e.,  $RT_{GPP}$  and  $RT_{ER}$ ). Color coding indicates whether metabolic activity was stimulated (blue), suppressed (red), or showed no significant change (yellow) in response to the storm event. The black line represents the linear regression between variables (only shown when p < 0.01) and the shaded area indicates the 95% confidence interval."

**Line 55/88: Clarify "recovery time" vs. River Network Saturation.**

Thanks, this distinction is important. As mentioned earlier, rather than indicating saturation of metabolic response, the recovery time pattern reflects a threshold in system resilience. We have added a clarification in the discussion (L389-391).

**Line 352: "ER recovery times at ca. 6 days (Fig. 5a)" -> it should be Fig. 5d**

We have fixed it as suggested

Diamond, J. S., Bernal, S., Boukra, A., Cohen, M. J., Lewis, D., Masson, M., Moatar, F., & Pinay, G.:Stream network variation in dissolved oxygen: Metabolism proxies and biogeochemical controls, *Ecological Indicators*, 131, 108233. https://doi.org/10.1016/j.ecolind.2021.108233, 2025.

Garcia, H. E. and Gordon, L. I.: Oxygen solubility in seawater: Better fitting equations, Limnology and Oceanography, 37, 1307–1312, https://doi.org/10.4319/lo.1992.37.6.1307, 1992.

Raymond, P. A., Zappa, C. J., Butman, D., Bott, T. L., Potter, J., Mulholland, P., Laursen, A. E., McDowell, W. H., and Newbold, D.: Scaling the gas transfer velocity and hydraulic geometry in streams and small rivers, *Limnology and Oceanography: Fluids and Environments*, 2, 41–53, https://doi.org/10.1215/21573689-1597669, 2012.

Odum, H. T.: Primary Production in Flowing Waters, *Limnology and Oceanography*, 1, 102–117, https://doi.org/10.4319/lo.1956.1.2.0102, 1956.

Wollheim, W. M., Bernal, S., Burns, D. A., Czuba, J. A., Driscoll, C. T., Hansen, A. T., Hensley, R. T., Hosen, J. D., Inamdar, S., Kaushal, S. S., Koenig, L. E., Lu, Y. H., Marzadri, A., Raymond, P. A., Scott, D., Stewart, R. J., Vidon, P. G., and Wohl, E.: River network saturation concept: factors influencing the balance of biogeochemical supply and demand of river networks, Biogeochemistry, 141, 503–521, https://doi.org/10.1007/s10533-018-0488-0, 2018.

---

## Author Response (AR2)

Dear editor and reviewers,

Thank you for your time and for the careful evaluation of our revised manuscript titled "Breathing Storms: Enhanced Ecosystem Respiration During Storms in a Heterotrophic Headwater Stream". We have followed all recommendations from both the reviewer and the editor and have revised the manuscript accordingly. All changes are indicated in the track-changed version, and below, we respond to each comment. The original text of the decision letter is in black and italics, while our responses are in dark blue.

General comments: The revised manuscript represents a substantial improvement over the initial submission.

The authors now report the frequency of high-flow days (less than 5% of storm days). The new Table S3 in the supplement is excellent. It explicitly links QC failures to discharge ranges and details the specific reasons for failure (e.g., poor model fit, implausible values). This directly addresses my request and improves the robustness of the paper.

The discussion appropriately states that the stopper hypothesis could not be empirically tested with reliable data from their study, and they provide plausible reasons for model failure at high flows (e.g., sensor burial). The updated discussion clearly states that no clear saturation pattern in ER was identified, contrasting with typical nutrient uptake patterns.

My last concern is: the re-analysis is described but not shown, making it unverifiable. The finding from this re-analysis show that 8 plausible ER estimates could be generated and that they did not show a stopper effect.

From your response letter: "For these days, high Q ranged from 100 to 1370 L s-1 and ER values were similarly elevated (range: 20 to 41g  $O_2$   $m^{-2}$   $d^{-1}$ ) as on days with large  $\Delta Q$  increases, providing no evidence for a metabolic stopper effect within our data".

--> Please provide the data from this re-analysis. A new supplementary table is required that lists the results for the 8 successful high-flow runs mentioned in your response letter.

We thank the reviewer for the positive feedback and constructive suggestions. In response, we have added a new supplementary table (Table S5) presenting the metabolic activity for all storm days with discharge greater than  $100 \text{ L s}^{-1}$ , including their quality check statistics (R2, RMSE). The table highlights both the unsuccessful runs and the 8 successful high-flow runs from the re-analysis with a constrained  $K_{600}$ .

Table S5. Results of the reanalysis of storm days with daily discharge above 100 L s-1 using *StreamMetabolizer* with constrained  $K_{600}$  to 100 d-1. For each date, the table shows daily discharge (Q), model fit metrics (R2, RMSE), and daily estimates of gross primary production (GPP, g O2 m-2 d-1), ecosystem respiration (ER, g O2 m-2 d-1), and reareation rates (K600, d-1). Values in bold indicate days that passed the quality check. Missing values (NA) indicate cases where reliable metabolic estimates could not be obtained..

| date       | Q                 | $\mathbb{R}^2$ | RMSE   | GPP                                              | ER                                                  | K600            |
|------------|-------------------|----------------|--------|--------------------------------------------------|-----------------------------------------------------|-----------------|
| dd/mm/yyyy | L s -1 |                |        | g O 2 m -2 d -1 | g O 2 m -2
d -1 | d -1 |
| 15/10/2018 | 157.2             | 98.40          | 6.74   | -0.03                                            | -20.34                                              | 99.99           |
| 01/11/2018 | 158.6             | 6.67           | 57.93  | 0.62                                             | -20.44                                              | 100.00          |
| 15/11/2018 | 104               | -54.44         | 69.32  | 0.19                                             | -20.63                                              | 100.00          |
| 16/11/2018 | 356.4             | -186.36        | 98.23  | -1.30                                            | -37.59                                              | 100.00          |
| 17/11/2018 | 218.8             | -16.32         | 61.90  | 0.53                                             | -33.13                                              | 100.00          |
| 18/11/2018 | 744.1             | -16009.87      | 714.99 | -3.95                                            | -67.99                                              | 99.86           |
| 19/11/2018 | 1037              | -2748.18       | 295.03 | 4.86                                             | -106.70                                             | 99.86           |
| 20/11/2018 | 283.2             | 80.42          | 24.05  | -1.85                                            | -54.05                                              | 100.00          |
| 21/11/2018 | 154.1             | -113.10        | 80.63  | 0.84                                             | -42.65                                              | 99.99           |
| 14/04/2020 | 193.2             | -223.85        | 105.56 | 1.25                                             | -39.60                                              | 99.98           |
| 15/04/2020 | 135               | -50.69         | 71.85  | 1.45                                             | -38.95                                              | 100.00          |
| 16/04/2020 | 108.8             | -222.15        | 104.70 | 1.30                                             | -35.49                                              | 99.99           |
| 19/04/2020 | 183.9             | -2341.77       | 284.57 | 0.50                                             | -34.16                                              | 99.97           |
| 20/04/2020 | 295.5             | -15.73         | 62.04  | 0.48                                             | -40.08                                              | 100.00          |
| 21/04/2020 | 2319.7            | -20530.77      | 828.47 | 0.77                                             | -51.19                                              | 99.81           |
| 22/04/2020 | 3883.7            | -3218.55       | 332.92 | 3.42                                             | -69.40                                              | 99.81           |
| 23/04/2020 | 364.8             | -1.19          | 57.75  | 0.33                                             | -48.12                                              | 100.00          |
| 24/04/2020 | 227.7             | 76.31          | 27.80  | 0.29                                             | -41.12                                              | 99.99           |
| 25/04/2020 | 169.1             | 96.06          | 11.23  | 0.05                                             | -38.53                                              | 99.98           |
| 26/04/2020 | 136.3             | 97.90          | 8.15   | 0.13                                             | -37.14                                              | 100.00          |
| 27/04/2020 | 115.3             | 98.25          | 7.39   | 0.06                                             | -36.86                                              | 100.00          |
| 21/03/2022 | 108.2             | NA             | NA     | NA                                               | NA                                                  | NA              |
| 28/04/2020 | 100.2             | 93.27          | 14.41  | 0.19                                             | -35.51                                              | 99.98           |
| 21/04/2022 | 259.5             | 64.28          | 35.16  | 0.49                                             | -36.35                                              | 99.96           |
| 22/04/2022 | 150.3             | 95.04          | 13.34  | 0.81                                             | -22.13                                              | 99.99           |
| 23/04/2022 | 104               | 74.67          | 29.90  | 0.61                                             | -19.69                                              | 99.99           |

Minor comments:

Line 25: ...was found for ER (R2 > 0.37)...recovery times were positively related to the size of the event only for ER (R2 > 0.46). The wording is slightly imprecise. Why not acknowledge the precise range/values?

In these two cases, changes in ER and recovery time of ER both exhibited a positive relationship with  $\Delta Q$ , with linear and logarithmic models providing equally strong fits based on AIC values. To maintain the abstract's conciseness and focus, we have chosen to report these results in a more general form, while the detailed model fits and values are presented in the main text.

L 20 (track-change version) "35 of them": Please rewrite it to indicate how you opted for 35 "selected" (?) storm events.

L 21 "considering all events": Do you mean "all selected 35 events"?

We have changed the sentence to "Due to data and model constraints, we were able to estimate metabolic rates for 35 of the events" to improve clarity. (L21)

L 23 "unrelated": Given the hydrologic theme of this study, this expression doesn't make sense logically. Please clarify whether you meant "not significantly related".

We have changed to "not statistically significant" as suggested. (L25)

L 51-55 & associated discussion (e.g., L 333-341; 360-365): Given the reviewer's concern about the "stopper" effect, I would invite you to revise the terms and definitions in line with the literature and other descriptions in the manuscript: For instance, refer to Fig. 5 in Covino, 2017 (http://dx.doi.org/10.1016/j.geomorph.2016.09.030), where high flows (connectivity) function as a kind of modulator or suppressor of reactivity, not as "stoppers". Unlike triggers or stoppers, I got the impression that the terms "stimulation" and "suppression" in Fig. I would not elicit any unnecessary misunderstanding regarding the often subtle, storm-induced metabolic shifts.

We have revised the terminology throughout the manuscript to align with terms used in the literature, including Covino's (2017) conceptual framework. Specifically, we replaced the terms "metabolic trigger" and "metabolic stopper" with the more process-oriented terms "stimulation" and "suppression" that are already used in our hypotheses (Fig. 1). We believe these revisions clarify the underlying processes, avoid misunderstandings, and ensure consistency with the hydrologic connectivity—reactivity framework described in the literature.

L 118 "are": Please use the past tense consistently when you describe methods and findings.

We have changed to "were" as suggested.

L 145 "DO": Please define this at its first use (L 131?) Given the importance of DO measurements in this study, it would be helpful if more details are provided, particularly with regard to sensor principle (optical or old membrane-type?) and maintenance (e.g., cleaning measures, especially to handle any sensor biofouling issue)

First use of "DO" in L49 has been defined. The MiniDOT used in this study is an optical (fluorescent optode) logger designed for robust, long-term field deployments with minimal maintenance requirements. To ensure data quality, sensors were inspected every 15–20 days for debris or biofilm, and the sensing surface and housing were gently rinsed with stream water when needed. In any case, note that sensors exhibited reduced biofouling due to oligotrophic water and low light inputs, which restricts the growth of organisms that form biofilms. These details have been added to the Methods section.

L 251: Just to double check whether r2 = 0.06 was significant at p < 0.001? By comparing the numbers described in L 251-253, I wondered about the significance of this marginal r2 number.

We appreciate the editor's attention to this point. We have double-checked the analysis and confirmed that the reported values are correct. Furthermore, we now specify in the manuscript that the relationship was weak, although statistically significant.

L 310: Please use "by" instead of "over".

We have changed to "by" as suggested. (L311)

L 387-390: Please rephrase or provide more supporting evidence to respond to the reviewer's concern.

We have revised the manuscript to better answer the reviewers' concern about the need to revisit the River Network Saturation concept and to differentiate the observed saturation in recovery time from the absence of saturation in the ER magnitude response. Updated text (L377-386).

"Moreover, the relationship between  $\Delta ER$  and  $\Delta Q$  was equally well explained by both linear and logarithmic models, indicating no statistically supported evidence for a saturation pattern in ER. This finding contrasts with the saturation responses often

reported for nutrient uptake with increasing discharge (sensu River Network Saturation concept; Wollheim et al., 2018). Given that the largest analysed storms fell within a moderate  $\Delta Q$  range compared to extreme hydrological events in similar systems, it is possible that the ecosystem's processing capacity was not fully exceeded, and thus the available range was insufficient to reveal a true saturation response. Nevertheless, we observed apparent constraints in ER during storms. For instance, ER rates were never below -36.4 g  $O_2$  m-2 d-1, and the maximum  $\Delta$ ER was around 100% indicating that storm-driven increases in ER never exceeded twice the prior baseflow rates. These observed constraints on the heterotrophic response to storm disturbances should be interpreted as empirical bounds within our dataset rather than definitive evidence of metabolic saturation in Fuirosos."

**L 424 "carbon cycling": Please articulate specific aspects of the aquatic carbon cycling related to your findings.**

We appreciate the reviewer's suggestion to be more specific regarding the aspects of aquatic carbon cycling related to our findings. We have revised the conclusion (L418–L423) to explicitly link our results to carbon processing pathways. The text now specifies that storm-driven stimulation of ER represents rapid pulses of organic carbon mineralization and CO2 emissions, and that the magnitude and frequency of these "breathing storms" influence both the total annual CO2 flux and its temporal variability, with potential effects on downstream C transport and emissions. We also clarify that changes in hydrological regimes could alter the timing, intensity, and cumulative magnitude of these pulses, thereby modifying key pathways of aquatic carbon cycling.